# CP-Guard+: A New Paradigm for Malicious Agent Detection and Defense in Collaborative Perception

## Abstract

Collaborative perception (CP) is a promising method for safe connected and autonomous driving, which enables multiple connected and autonomous vehicles (CAVs) to share sensing information with each other to enhance perception performance. For example, occluded objects can be detected, and the sensing range can be extended. However, compared with single-agent perception, the openness of a CP system makes it more vulnerable to malicious agents and attackers, who can inject malicious information to mislead the perception of an ego CAV, resulting in severe risks for the safety of autonomous driving systems. To mitigate the vulnerability of CP systems, we first propose a new paradigm for malicious agent detection that effectively identifies malicious agents at the feature level without requiring verification of final perception results, significantly reducing computational overhead. Building on this paradigm, we introduce CP-GuardBench, the first comprehensive dataset provided to train and evaluate various malicious agent detection methods for CP systems. Furthermore, we develop a robust defense method called CP-Guard+, which enhances the margin between the representations of benign and malicious features through a carefully designed mixed contrastive training strategy. Finally, we conduct extensive experiments on both CP-GuardBench and V2X-Sim, and the results demonstrate the superiority of CP-Guard+.

## 1 Introduction

The development of collaborative perception (CP) has been driven by the increasing demand for accurate and reliable perception in autonomous driving systems (Chen et al., 2019b;a; Li et al., 2022; Hu et al., 2024d;a; 2023; Fang et al., Aug. 2024; Xu et al., 2022). Single-agent perception systems, which rely solely on the onboard sensors of a single CAV, are restricted by limited sensing range and occlusion. On the contrary, CP systems incorporate multiple CAVs to collaboratively capture their surrounding environments. Specifically, The CAVs in a CP system can be divided into two categories: the ego CAV and helping CAVs. The helping CAVs send complementary sensing information (most methods send intermediate features) to the ego CAV, and the ego CAV then leverages this complementary information to enhance its perception performance (Balkus et al., 2022; Han et al., 2023; Hu et al., 2024c; Wang et al., 2020). For example, the ego CAV can detect occluded objects and extend the sensing range after fusing the received information.

Despite the many advantages of CP outlined above, it also has several crucial drawbacks. Compared to single-agent perception systems, CP is more vulnerable to security threats and easier to be attacked, since it requires receiving and fusing information from other CAVs, which expands the attack surface. In particular, malicious agents can directly send intermediate features with adversarial perturbations to fool the ego CAV or a man-in-the-middle who can capture the intermediate feature maps and manipulate them. Figure 1a illustrates the vulnerability of CP to malicious agents. In addition, several attack methods have been designed to fool CP. For example, Tu *et al.* (Tu et al., 2021) developed a method to generate indistinguishable adversarial perturbations to attack the multi-agent communication in CP, which can severely degrade the perception performance.

The inability of the ego CAV to accurately detect and eliminate malicious agents from its collaboration network poses significant risks to CP, potentially resulting in compromised perception outcomes

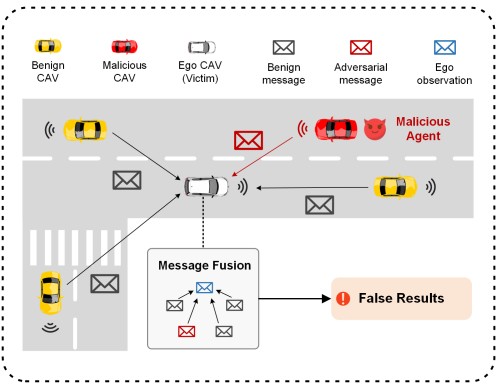

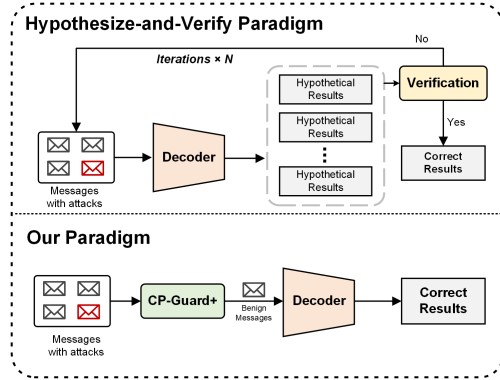

(a) Threats in Collaborative Perception (b) Paradigm Comparison

Figure 1: (a) **Illustration of the threats of malicious agent in collaborative perception**. Malicious CAVs could send intricately crafted adversarial messages to an ego CAV, which will mislead it to generate false positive perception outputs. (b) **Comparison between the proposed CP-Guard+ with the traditional hypothesize-and-verify malicious agent detection methods.** Hypothesize-and-verify involves multiple rounds of malicious agent detection iterations at the output level, and requires the generation of multiple hypothetical outputs for verification, incurring high computational overhead. In contrast, CP-Guard+ directly outputs robust CP results with intermediate feature-level detection, significantly reducing the computational overhead.

and catastrophic consequences. For instance, the ego CAV might misinterpret traffic light statuses or fail to detect objects ahead of the road, resulting in severe traffic accidents or even fatalities. Hence, it is crucial to develop a defense mechanism for CP that is resilient to attacks from malicious agents and capable of eliminating them from its collaboration network.

To address the security threats in CP, some previous works have investigated the defense mechanisms against malicious agents. For example, Li *et al.* (Li et al., 2023) leveraged random sample consensus (RANSAC) to sample a subset of collaborators and calculate the intersection of union (IoU) of the bounding boxes to verify whether there is any malicious agent among the collaboration network. Zhao *et al.* (Zhao et al., 2024) designed a match loss and a reconstruction loss as statistics to measure the consensus between the ego CAV and the collaborators. In addition, our previous work, CP-Guard, which is currently under review, defends against malicious agents by iteratively checking the anomaly of the collaborative segmentation results from different collaborators. However, these methods all follow a hypothesize-and-verify paradigm, which requires generating multiple hypothetical perception results and verifying the consistency between the ego CAV and the collaborators. This process is computation-intensive and time-consuming, which hinders its scalability. This limitation prompts us to explore a new paradigm:

*Is it feasible to detect malicious agents directly at the feature level?*

As illustrated in Figure 1b, the new paradigm shifts the focus to feature-level detection, eliminating the need to generate multiple hypothetical perception results. This direct approach can significantly reduce the computational overhead, thereby enhancing the efficiency of malicious agent detection in CP systems.

Although this idea is concise and appealing, there are still some challenges in realizing it. Firstly, to detect malicious agents at the feature level, we need to train a deep neural network (DNN) model on a large-scale dataset to help it learn the features of benign and malicious agents. However, there is a lack of a benchmark dataset for feature-level malicious agent detection in CP systems. The existing datasets for CP, such as V2X-Sim (Li et al., 2022) and OPV2V (Xu et al., 2022), contain only benign agents and do not include malicious agents. Therefore, it is difficult to train a robust DNN model for malicious agent detection in CP systems without a well-annotated dataset. Secondly, in CP scenarios, the environments are highly dynamic and complex, making it unrealizable to directly use a classifier to classify the received intermediate features for detecting malicious agents. This is because dynamic environments will cause a high false-positive rate (FPR). Additionally, the adver-

sarial perturbations are indistinguishable at the feature level, and the feature distribution of malicious agents and benign agents are highly similar. These factors make it difficult to train a robust model to distinguish malicious agents from benign agents.

To address the aforementioned challenges, we first generate a new dataset, CP-GuardBench, which is the first dataset for malicious agent detection in CP systems. Then, we propose CP-Guard+, a robust malicious agent detection method for CP systems. CP-Guard+ can effectively detect malicious agents at the feature level without verifying the final perception results, significantly reducing computational overhead and enhancing defense efficiency. Moreover, we design a mixed contrastive training strategy to tackle the stealthy challenges and further enhance the robustness.

In summary, we investigate the malicious agent detection problem in CP systems and propose a brand new paradigm, feature-level malicious agent detection. Additionally, we construct CP-GuardBench, the first benchmark for malicious agent detection in CP systems. Furthermore, we propose CP-Guard+, a robust malicious agent detection method with high robustness and computational efficiency. Finally, we conduct extensive experiments on CP-GuardBench and V2X-Sim, and the results demonstrate the superiority of our CP-Guard+.

## 2 PRELIMINARIES

### 2.1 FORMULATION OF COLLABORATIVE PERCEPTION

In this section, we formulate collaborative perception and give the pipeline of our CP system. Specifically, Let $\mathcal{X}^N$ denote the set of $N$ CAVs in the CP system. CAVs in $\mathcal{X}$ can be divided into two categories: the ego CAV and helping CAVs. The ego CAV is the one that needs to perceive its surrounding environment, while helping CAVs are the ones that send their complementary sensing information to the ego CAV to help it enhance its perception performance. Thus, each CAV can be an ego one and helping one, depending on its role in a perception process. We assume that each CAV is equipped with a feature encoder $f_{\text{encoder}}(\cdot)$, a feature aggregator $f_{\text{aggregator}}(\cdot)$, and a feature decoder $f_{\text{decoder}}(\cdot)$. For the $i$-th CAV in the set $\mathcal{X}$, the raw observation is denoted as $\mathbf{O}_i$ (such as camera images and LiDAR point clouds), and the final perception results are denoted as $\mathbf{Y}_i$. The CP pipeline of the $i$-th CAV can be described as follows.

1. *Observation Encoding*: Each CAV encodes its raw observation $\mathbf{O}_j$ into an initial feature map $\mathbf{F}_j = f_{\text{encoder}}(\mathbf{O}_j)$, where $j \in \mathcal{X}^N$.

2. *Intermediate Feature Transmission*: Helping CAVs transmit their intermediate features to the ego CAV: $\mathbf{F}_{j \to i} = \mathbf{\Gamma}_{j \to i}(\mathbf{F}_j)$, $j \in \mathcal{X}^N, j \neq i$, where $\mathbf{\Gamma}_{j \to i}(\cdot)$ denotes a transmitter that conveys the $j$-th CAV's intermediate feature $\mathbf{F}_j$ to the ego CAV, while performing a spatial transformation. $\mathbf{F}_{j \to i}$ is the spatially aligned feature in the $i$-th CAV's coordinate.

3. *Feature Aggregation*: The ego CAV receives all the intermediate features and fuses them into a unified observational feature $\mathbf{F}_{\text{fused}} = f_{\text{aggregator}}(\mathbf{F}_{0 \to i}, \{\mathbf{F}_{j \to i}\}_{j \neq i, \ j \in \mathcal{X}^N})$.

4. *Perception Decoding*: Finally, the ego CAV decodes the unified observational feature $\mathbf{F}_{\text{fused}}$ into the final perception results $\mathbf{Y} = f_{\text{decoder}}(\mathbf{F}_{\text{fused}})$.

### 2.2 ADVERSARIAL THREAT MODEL

Our focus is on the intermediate-fusion collaboration scheme, where an attacker introduces designed adversarial perturbations on the intermediate features to subtly mislead the perception of the ego CAV. Since the attacker installs the perception model locally to participate in the collaborative system, we assume they have white-box access to the model parameters. The attack procedure can be formulated as follows.

$$\mathbf{F}_k = f_{\text{encoder}}(\mathbf{O}_k), \quad k \in \mathcal{X}^N, \tag{1}$$

$$\mathbf{F}_k^{\delta} = \mathbf{F}_k + \delta, \tag{2}$$

$$\mathbf{F}_{k \to i}^{\delta} = \mathbf{\Gamma}_{k \to i}(\mathbf{F}_k^{\delta}), \quad k \in \mathcal{X}^N, k \neq i, \tag{3}$$

$$\mathbf{F}_{\text{fused}}^{\delta} = f_{\text{aggregator}}(\mathbf{F}_{0 \to i}, \mathbf{F}_{k \to i}^{\delta}, \{\mathbf{F}_{j \to i}^{\delta}\}_{j \neq i, \ j \in \mathcal{X}^N}), \tag{4}$$

$$\mathbf{Y}^{\delta} = f_{\text{decoder}}(\mathbf{F}_{\text{fused}}^{\delta}), \tag{5}$$

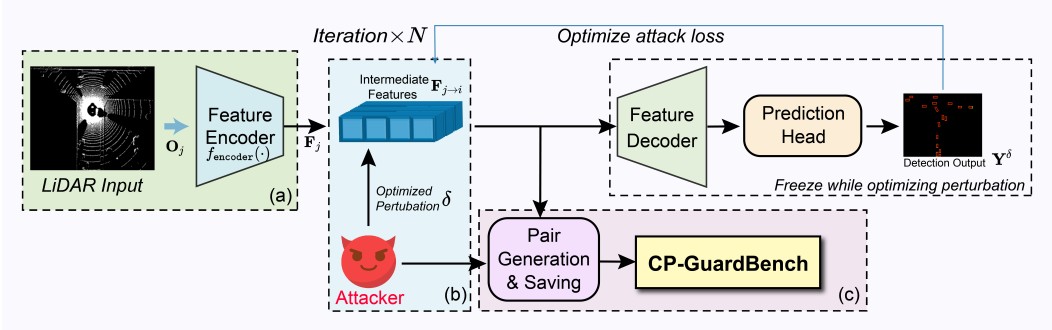

Figure 2: **Automatic Data Generation and Annotation Pipeline.** We first train a robust LiDAR collaborative object detector. Then, we discard the detection head and decoder, and only keep the backbone as the intermediate feature generator. The data generation pipeline is shown in (a), (b), and (c), where (a) is the intermediate feature generation, (b) is the attack implementation, and (c) is the pair generation and saving.

where $k$-th agent is malicious, and $\delta$ denotes the adversarial perturbation generated by the attacker. $i$-th agent is the ego CAV. In addition, the attacker's objective is to optimize the adversarial perturbation $\delta$ to maximize the loss function of the ego CAV. The optimization problem can be formulated as follows.

$$\arg\max_{\delta} \mathcal{L}(\mathbf{Y}^\delta, \mathbf{Y}^{\mathtt{gt}}), \quad \mathtt{s.t.} \quad \|\delta\| \leq \Delta \tag{6}$$

where $\mathcal{L}(\cdot)$ denotes the loss function, $\mathbf{Y}^\delta$ is the attacked CP results obtained from Eq. 5, and $\mathbf{Y}^{\mathtt{gt}}$ is the ground truth. Pertubation $\delta$ is constrained by $\|\delta\| \leq \Delta$ to ensure its stealth to avoid being detected. Moreover, as for the physical sensor attacks, such as LiDAR or GPS spoofing, we do not consider them in this study, as these are general threats to CAVs, and our focus is on vulnerabilities specific to CP. Furthermore, we assume that the attacker cannot bypass cryptographic protections, thereby preserving the security of communication channels between vehicles.

## 3 CP-GUARDBENCH

To facilitate feature-level malicious agent detection in CP systems, we propose to develop CP-GuardBench, the first benchmark for malicious agent detection in CP systems. It provides a comprehensive dataset for training and evaluating malicious agent detection methods. In this section, we will introduce the details of CP-GuardBench, including the automatic data generation and annotation pipeline in Section 3.1, and the data visualization and statistics in Section 3.2.

### 3.1 AUTOMATIC DATA GENERATION AND ANNOTATION

We build CP-GuardBench based on one of the most widely used datasets in the CP field, V2X-Sim (Li et al., 2022), which is a comprehensive simulated multi-agent perception dataset for V2X-aided autonomous driving. In this section, we introduce the automatic data generation and annotation pipeline of CP-GuardBench. The pipeline is shown in Figure 2. It consists of three steps: 1) intermediate feature generation, 2) attack implementation, and 3) pair generation and saving.

Specifically, we firstly train a robust LiDAR collaborative object detector, which consists of a convolutional backbone, a convolutional decoder, and a prediction head for classification and regression (Luo et al., 2018). As for the fusion method, we adopt mean fusion method to fuse the intermediate features from different collaborators. Subsequently, the backbone is retained for extracting intermediate features, which are then transmitted and utilized by an ego CAV as supplementary information.

Secondly, the attacks are implemented and applied to the intermediate features. The detection head and decoder are then frozen to generate the attacked detection results and optimize the adversarial perturbations. As shown in Figure 2, several iterations are required to optimize the perturbations, and the loss function differs for different attack types. In our CP-GuardBench, we consider five types of attacks, including Projected Gradient Descent (PGD) (Madry et al., 2018), Carini & Wagner (C&W)

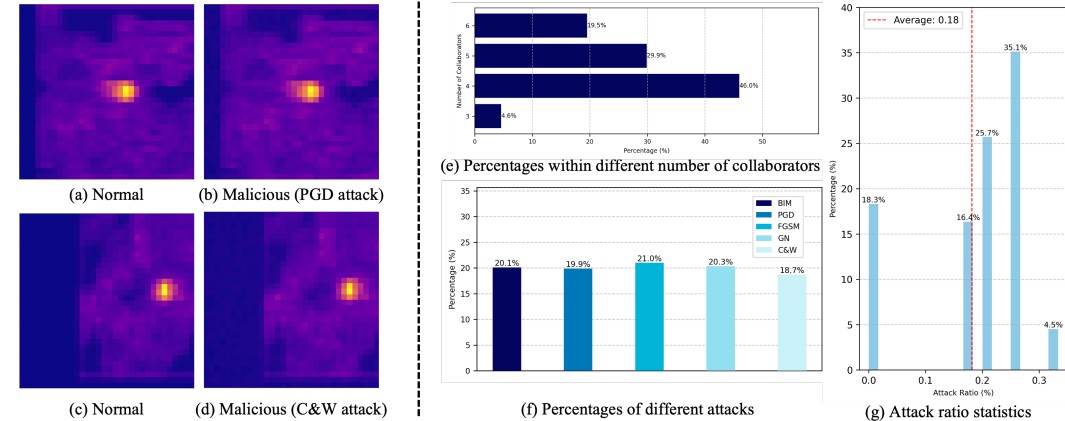

Figure 3: **Visualization and Statistics of CP-GuardBench. (a), (b), (c) and (d) are visualization**, which visualize the normal intermediate features and the adversarial examples perturbed by different malicious agents. We can see the adversarial examples are almost identical to the normal examples, which indicates the challenges in detecting malicious agents. **(e), (f), (g) and (h) are the statistics** of CP-GuardBench, including the number of collaborators, attack ratio and attack types.

attack (Carlini & Wagner, 2017), Basic Iterative Method (BIM) (Kurakin et al., 2017), Guassian Noise Perturbation (GN), and Fast Gradient Sign Method (FGSM) (Goodfellow et al., 2015). The implementation details can be found in the Appendix C.

In the generation of attack data, we randomly choose one of the attacks above and generate the corresponding attack data in each iteration. Finally, the perturbed features will be annotated with the corresponding attack type and saved for later use.

### 3.2 DATA VISUALIZATION AND STATISTICS

We visualize the samples of the generated data in Figures 3 (a), (b), (c) and (d). We observe that the attacks are so stealthy that it is very hard to see the difference with the naked eye, which poses a great challenge to address the malicious agent detection.

To construct CP-GuardBench, we randomly sample 9000 frames from V2X-Sim and generate 42200 feature-label pairs. The data is then split into training, validation, and test sets with a ratio of 8:1:1. The data statistics are shown in Figures 3 (e), (f), and (g). Figure 3 (e) illustrates the distribution of the number of collaborators, which is the number of agents that collaboratively perceive the environments. The number of collaborators ranges from 3 to 6, with the most common scenario being 4 collaborators, accounting for 46.0% of the total data. 5 and 6 collaborators are also common, accounting for 29.9% and 19.5% of the total data, respectively. Regarding the distribution of attack types, as depicted in Figure 3 (f), we observe that the attack types are evenly distributed, with each type accounting for approximately 20% of the total data. This is due to the random selection of one attack type in each iteration. Figure 3 (g) illustrates the attack ratio, which represents the ratio of the number of attackers to the total number of agents in a collaboration network. The maximum attack ratio exceeds 0.3, the minimum is 0, and the average attack ratio is 0.18.

## 4 CP-GUARD+

### 4.1 RESIDUAL LATENT FEATURE LEARNING

As discussed in Section 1, the detection of malicious agents in CP scenarios at the feature level is a challenging task due to the highly dynamic nature of the environments. This dynamism leads to non-stationary data distributions with significant noise. If a model is directly used to detect malicious agents, it may not always accurately estimate the latent distribution, particularly when the input is too noisy to perform effective dimension reduction. For instance, object detectors often have feature maps that include complex information from both the foreground objects and the noisy background before aggregation.

Figure 4: **Illustration of the Mixed Contrastive Training.** In each iteration, we generate a batch of pairs and the features of benign agents and malicious agents are projected to one-dimensional vectors. After the mixed contrastive training, the features of benign agents and malicious agents are regularized to respectively cluster to a compact space and reduce the overlap between the two spaces.

To address this challenge, we propose a residual latent feature learning mechanism, which means we do not learn the features of the benign or malicious agents' intermediate feature maps directly. Instead, we learn the residual features of the collaborator's feature maps with respect to the ego agent's feature maps. This way, the model can focus on the differences between the benign and malicious agents' feature maps.

This mechanism is also inspired by the idea that the collaborators' intermediate feature maps will achieve a consensus rather than a conflict against the ego CAV's intermediate feature maps. In other words, we can learn the residual latent feature between a CAV and its corresponding collaborators to check the consensus between them.

Specifically, consider the collaborators' intermediate feature maps $\{\mathbf{F}_{j \to i}\}_{j \neq i, \ j \in \mathcal{X}^N}$ and the ego CAV's intermediate feature maps $\mathbf{F}_i$. We can obtain the residual latent feature by

$$\mathbf{F}_{j \to i}^{\texttt{res}} = \mathbf{F}_i - \mathbf{F}_{j \to i}. \tag{7}$$

Then, we can leverage the residual latent feature to detect malicious agents by modeling the detection problem as a binary classification task. A binary classifier $f_{\texttt{classifier}}(\mathbf{x}; \theta)$ is trained on the residual latent feature to distinguish between benign (labeled 0) and malicious (labeled 1) agents. The model is optimized using the cross-entropy loss, as defined below.

$$\mathcal{L}_{\texttt{res}} = -\frac{1}{N} \sum_{i=1}^{N} \left( y_i \log(p_i) + (1 - y_i) \log(1 - p_i) \right), \tag{8}$$

where $N$ is the number of samples, $y_i$ is the ground truth label, $p_i$ is the predicted probability, and $\theta$ is the classifier's parameters that need to be optimized.

### 4.2 MIXED CONTRASTIVE TRAINING

The attacker can generate adversarial perturbations that are indistinguishable from the benign features, and the feature distribution of malicious agents and benign agents is highly similar, which makes it difficult to train a robust model to distinguish malicious agents from benign agents. To address this challenge, we propose a mixed contrastive training strategy to enhance the robustness of the model. The idea is to regularize the benign and malicious features of these data to respectively cluster to a compact space regardless of their distributions while reducing the clusters' overlap. This is crucial. If we use traditional training strategies, the model might fail to project the residual latent features with class-specific cohesion and separation that is independent of the distribution, which can lead to ambiguous predictions and increased sensitivity to distribution shifts.

Specifically, consider the benign and malicious intermediate feature maps, $\mathbf{F}_{j \to i}$ and $\mathbf{F}_{k \to i}^{\delta}$, which have been transmitted and spatially aligned with the ego CAV. We can obtain the residual latent feature by Eq. 7, $\mathbf{F}_{j \to i}^{\texttt{res}}$ and $\mathbf{F}_{k \to i}^{\texttt{res}, \delta}$, respectively. After that, we use a multi-layer perceptron (MLP) to project the residual latent features into one-dimensional vectors $\{\mathcal{V}_i\}_{i=0,1,\dots,N}$.

Then, we enhance the distinctiveness of these features by ensuring that they are closely grouped within the same class and well separated from different classes, regardless of their distribution. Here, we leverage the InfoNCE (Chen et al., 2020) objective to impose such regularization. Denote $(\mathcal{V}_m, \mathcal{V}_n)$ as a pair of features, which is a positive pair if they are from the same class (both benign or malicious) and a negative pair otherwise.

$$\ell\left(\mathcal{V}_m, \mathcal{V}_n\right) = -\log \frac{\exp\left(\mathcal{V}_m \odot \mathcal{V}_n / \tau\right)}{\sum_{o=1, o \neq m}^{N} \mathbb{I}\left(\mathcal{V}_m, \mathcal{V}_o\right) \cdot \exp\left(\mathcal{V}_m \odot \mathcal{V}_o / \tau\right)} \tag{9}$$

where $\mathbb{I}(\mathcal{V}_m, \mathcal{V}_o)$ is an indicator function that returns one or zero for positive and negative pairs, respectively. $\tau$ is a temperature parameter and $\odot$ denotes the cosine similarity, where $\mathcal{V}_m \odot \mathcal{V}_n = \frac{\mathcal{V}_m^\top \mathcal{V}_n}{\|\mathcal{V}_m\|\|\mathcal{V}_n\|}$. The final objective function is the average of $\ell$ over all positive pairs.

$$\mathcal{L}_{\texttt{ctrs}} = \frac{1}{C(N, 2)} \sum_{m=1}^{N} \sum_{n=m+1}^{N} \left(1 - \mathbb{F}(\mathcal{V}_m, \mathcal{V}_n)\right) \cdot \ell\left(\mathcal{V}_m, \mathcal{V}_n\right) \tag{10}$$

where $C(N, 2) = \binom{N}{2} = \frac{N!}{2!(N-2)!}$. During training, we use the combination of Eq. 10 and Eq. 8 to optimize the model:

$$\mathcal{L}_{\texttt{mixed}} = \mathcal{L}_{\texttt{res}} + \alpha \cdot \mathcal{L}_{\texttt{ctrs}} \tag{11}$$

where $\alpha$ is a hyperparameter to balance the two losses. By doing so, the first term $\mathcal{L}_{\texttt{res}}$ quantifies the difference between the true distribution and the predicted distribution from the model, thereby penalizing the confidence in wrong predictions. More importantly, as shown in Figure 4, the second term $\mathcal{L}_{\texttt{ctrs}}$ regularizes the features of benign agents and malicious agents to cluster into compact spaces and reduces the overlap between the two spaces, which is important for the model to learn the residual latent features with class-specific cohesion and separation. This strategy makes the model more robust to the distribution overlap and yield better performance on malicious agent detection.

# 5 EXPERIMENTS

## 5.1 EXPERIMENTAL SETUP

**Dataset and Baselines.** In our experiments, we consider two datasets: CP-GuardBench and V2X-Sim (Li et al., 2022). We designate CAV #1 as the ego CAV and randomly select adversarial collaborators from the remaining CAVs. Additionally, we use ROBOSAC (Li et al., 2023) and MADE (Zhao et al., 2024) as baselines, which are two state-of-the-art CP defense methods based on the hypothesize-and-verify paradigm.

**Attack Settings.** We assess different CP defense methods targeted at five attacks: PGD attack (Madry et al., 2018), C&W attack (Carlini & Wagner, 2017), BIM attack (Kurakin et al., 2017), FGSM attack (Goodfellow et al., 2015), and GN attack. We set different perturbation sizes $\Delta \in \{0.1, 0.25, 0.5, 0.75, 1.0\}$. The number of malicious attackers varies in $\{0, 1, 2\}$ and all the attackers are randomly assigned from the collaborators, where 0 attacker indicates an upper-bound case. For PGD, BIM and C&W attacks, the number of iteration steps is 15 and the step size is 0.1.

**Implementation Details.** The CP-Guard+ system is implemented using PyTorch, and we utilize the object detector described in Section 3.1. For each agent, the local LiDAR point cloud data is first encoded into $32 \times 32$ bird's eye view (BEV) feature maps with 256 channels prior to communication. For our CP-Guard+, we use ResNet-50 (He et al., 2016) as the backbone, and the training is performed for 50 epochs with batch size 10 and learning rate $1 \times 10^{-3}$. Our experiments are conducted on a server with 2 Intel(R) Xeon(R) Silver 4410Y CPUs (2.0GHz), 4 NVIDIA RTX A5000 GPUs, and 512 GB DDR4 RAM. For mixed contrastive training, we utilize the output of the fully connected layers preceding the final output layer in the backbone to form a one-dimensional feature vector for each agent, the dimension of which is 2048.

**Evaluation Metrics.** We use a variety of metrics to evaluate the performance of our CP-Guard+ model. For malicious agent detection on our CP-GuardBench dataset, we consider Accuracy, True Positive Rate (TPR), False Positive Rate (FPR), Precision, and F1 Score. For CP defense on the V2X-Sim dataset, we use metrics including average precision (AP) at IoU=0.5 and IoU=0.7. Additionally, to assess the computation efficiency of different CP defense methods, we introduce the metric frames-per-second (FPS). The definition of the metrics are provided in Table 3 in Appendix.

Table 1: **Performance Evaluation of CP-Guard+ on CP-GuardBench.** We report the average accuracy, true positive rate (TPR), false positive rate (FPR), precision, and F1 score of CP-Guard+ on CP-GuardBench with different attack methods and perturbation budgets $\Delta = 0.25, 0.5, 0.75, 1.0$.

| Metrics | $\Delta = 0.25$ | | | | | $\Delta = 0.5$ | | | | |
| | Accuracy ↑ | TPR ↓ | FPR ↓ | Precision ↑ | F1 Score ↑ | Accuracy ↑ | TPR ↑ | FPR ↓ | Precision ↑ | F1 Score ↑ |
|---|---|---|---|---|---|---|---|---|---|---|
| PGD | 98.77 | 100.00 | 1.54 | 94.19 | 97.01 | 98.83 | 100.00 | 1.46 | 94.52 | 97.18 |
| BIM | 98.90 | 100.00 | 1.37 | 94.81 | 97.33 | 98.90 | 100.00 | 1.37 | 94.79 | 97.32 |
| C&W | 98.60 | 99.30 | 1.58 | 94.02 | 96.59 | 97.96 | 100.00 | 2.56 | 90.38 | 95.19 |
| FGSM | 91.64 | 64.41 | 1.46 | 91.79 | 75.70 | 97.53 | 93.22 | 1.37 | 94.50 | 93.86 |
| GN | 90.95 | 60.34 | 1.29 | 92.23 | 72.95 | 97.19 | 92.12 | 1.54 | 93.73 | 92.92 |
| Average | 95.78 | 84.81 | 1.44 | 93.43 | 87.93 | 98.08 | 97.07 | 1.66 | 93.45 | 95.29 |

| Metrics | $\Delta = 0.75$ | | | | | $\Delta = 1.0$ | | | | |
| | Accuracy ↑ | TPR ↑ | FPR ↓ | Precision ↑ | F1 Score ↑ | Accuracy ↑ | TPR ↑ | FPR ↓ | Precision ↑ | F1 Score ↑ |
|---|---|---|---|---|---|---|---|---|---|---|
| PGD | 98.83 | 100.00 | 1.46 | 94.55 | 97.20 | 98.60 | 100.00 | 1.75 | 93.50 | 96.64 |
| BIM | 98.90 | 100.00 | 1.37 | 94.81 | 97.33 | 98.66 | 100.00 | 1.67 | 93.73 | 96.76 |
| C&W | 97.30 | 100.00 | 3.41 | 88.46 | 93.88 | 96.43 | 100.00 | 4.42 | 84.34 | 91.50 |
| FGSM | 98.63 | 98.63 | 1.37 | 94.75 | 96.66 | 98.83 | 100.00 | 1.46 | 94.48 | 97.16 |
| GN | 98.49 | 98.27 | 1.45 | 94.35 | 96.27 | 98.90 | 99.96 | 1.28 | 95.08 | 97.32 |
| Average | 98.43 | 99.38 | 1.81 | 93.38 | 96,27 | 98.53 | 99.93 | 1.82 | 93.20 | 96.43 |

Table 2: **Comparative results of CP-Guard+ on V2X-Sim Dataset.** We report the AP@0.5 and AP@0.7 with different perturbation budgets $\Delta$ and number of malicious agents $N_{mal}$.

| Method | $\Delta = 0.25, N_{mal} = 1$ | | $\Delta = 0.5, N_{mal} = 1$ | | $\Delta = 0.25, N_{mal} = 2$ | | $\Delta = 0.5, N_{mal} = 2$ | |
| | AP@0.5 | AP@0.7 | AP@0.5 | AP@0.7 | AP@0.5 | AP@0.7 | AP@0.5 | AP@0.7 |
|---|---|---|---|---|---|---|---|---|
| Upper-bound | 79.94 | 78.40 | 79.94 | 78.40 | 79.94 | 78.40 | 79.94 | 78.40 |
| MADE (against PGD attack) | 64.63 | 45.22 | 64.81 | 44.89 | 62.45 | 43.49 | 63.04 | 43.77 |
| MADE (against C&W attack) | 65.26 | 45.24 | 64.74 | 45.65 | 63.41 | 44.28 | 62.86 | 42.93 |
| MADE (against BIM attack) | 66.11 | 45.94 | 65.51 | 45.47 | 64.36 | 43.89 | 63.56 | 44.09 |
| MADE Average | 65.33 | 45.47 | 65.02 | 45.34 | 63.41 | 43.89 | 63.15 | 43.60 |
| ROBOSAC (against PGD attack) | 62.13 | 42.90 | 63.67 | 43.79 | 59.01 | 40.03 | 59.97 | 40.44 |
| ROBOSAC (against C&W attack) | 61.83 | 42.01 | 62.47 | 42.80 | 59.39 | 39.94 | 59.83 | 39.82 |
| ROBOSAC (against BIM attack) | 62.69 | 43.80 | 63.78 | 43.66 | 59.10 | 39.74 | 59.29 | 39.89 |
| ROBOSAC Average | 62.21 | 42.90 | 63.31 | 43.42 | 59.37 | 39.90 | 59.70 | 40.05 |
| CP-Guard+ (against PGD attack) | 72.89 | 71.45 | 69.50 | 68.56 | 69.50 | 67.92 | 66.09 | 64.82 |
| CP-Guard+ (against C&W attack) | 69.41 | 66.86 | 60.64 | 55.41 | 64.17 | 61.73 | 58.54 | 53.15 |
| CP-Guard+ (against BIM attack) | 73.35 | 71.46 | 66.83 | 66.05 | 70.91 | 69.11 | 66.30 | 64.62 |
| CP-Guard+ Average | 71.88 | 69.92 | 65.66 | 63.34 | 68.19 | 66.25 | 63.64 | 60.86 |
| No Defense (PGD attack) | 29.73 | 28.47 | 11.35 | 11.17 | 12.69 | 12.42 | 1.69 | 1.65 |
| No Defense (C&W attack) | 19.03 | 16.58 | 4.69 | 3.78 | 19.03 | 16.58 | 0.71 | 0.58 |
| No Defense (BIM attack) | 26.69 | 25.71 | 10.05 | 9.89 | 11.59 | 11.38 | 1.37 | 1.33 |
| No Defense Average | 25.15 | 23.59 | 8.70 | 8.28 | 14.44 | 13.46 | 1.27 | 1.19 |

## 5.2 QUANTITATIVE RESULTS

**Performance Evaluation of CP-Guard+.** We test our CP-Guard+ model on the CP-GuardBench dataset under various attack methods and perturbation budgets ($\Delta$), as shown in Table 1. The metrics considered include Accuracy, True Positive Rate (TPR), False Positive Rate (FPR), Precision, and F1 Score. For a perturbation budget of $\Delta = 0.25$, CP-Guard+ achieves high accuracy across all attack methods, with PGD, BIM, and C&W attacks showing accuracy above 98%. FGSM and GN attacks result in relatively lower accuracy, around 91.64% and 90.95%, respectively. This is reasonable since these two attacks are weaker than other attacks and do not cause severe damage to the model unless there is a large perturbation. As the perturbation budget increases to $\Delta = 0.5$, the model maintains high performance, with an average accuracy of 98.08% and a TPR of 97.07%. For higher perturbation budgets ($\Delta = 0.75$ and $\Delta = 1.0$), the model continues to perform well, achieving an average accuracy of 98.43% and 98.53%, respectively. Notably, the TPR remains high across all perturbation budgets, indicating the model's robustness in detecting true positives. The FPR remains low, further demonstrating the model's effectiveness in minimizing false positives. Overall, CP-Guard+ exhibits strong performance and resilience against various attack methods and perturbation levels, maintaining high accuracy, precision, and F1 scores.

**Performance Comparison with Other Defenses.** We further compare AP@0.5 and AP@0.7 of our CP-Guard+ with other CP defense methods on the V2X-Sim dataset, including MADE (Zhao et al., 2024) and ROBOSAC (Li et al., 2023). As depicted in Table 2, when there is no defense against CP attacks, there is a significant drop in AP@0.5/0.7 compared to the upper-bound case. Moreover, increasing either the number of malicious agents or the perturbation level can lead to a further decline in AP. In contrast, taking measures to recognize malicious agents in advance can

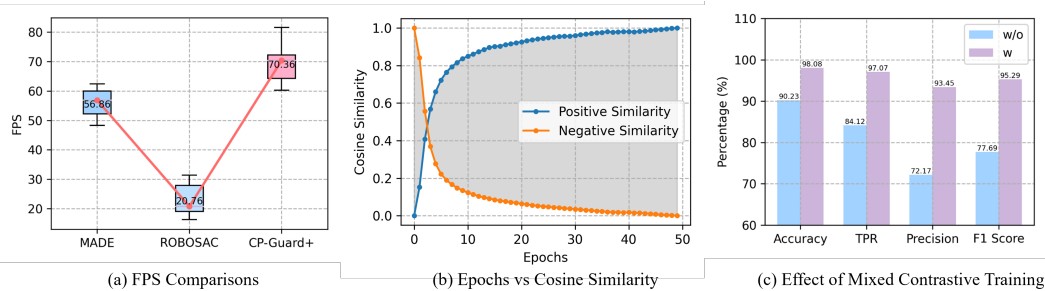

(a) FPS Comparisons     (b) Epochs vs Cosine Similarity     (c) Effect of Mixed Contrastive Training

Figure 5: **(a) FPS performance comparison** between CP-Guard+ with and other baselines. **(b) Cosine disctance** between the intermediate features of the malicious agent and the benign agent. **(c) Abalation study on the effectiveness of the mixed contrastive training**. 'w/o' means the CP-Guard+ *without the mixed contrastive training*. 'w/' means the CP-Guard+ *with the mixed contrastive training*.

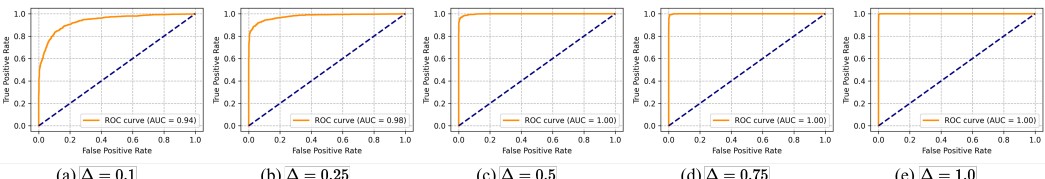

(a) $\Delta = 0.1$     (b) $\Delta = 0.25$     (c) $\Delta = 0.5$     (d) $\Delta = 0.75$     (e) $\Delta = 1.0$

Figure 6: ROC curve of CP-Guard+ on CP-GuardBench.

effectively prevent CP performance degradation, with our CP-Guard+ showing the highest scores. For attacks with $\Delta = 0.25$ and $N_{mal} = 1$, our CP-Guard+ achieves an average of 71.88% AP@0.5 and 69.92% AP@0.7 against three attacks, which are 186.81% and 196.40% higher than the no-defense case, respectively. Compared to MADE, CP-Guard+ achieves 10.03% and 53.77% higher AP@0.5 and AP@0.7, respectively. For ROBOSAC, our CP-Guard+ achieves 15.54% and 62.98% higher AP@0.5 and AP@0.7, respectively. These results highlight the superiority of our CP-Guard+ over existing CP defense methods. Additionally, as the number of malicious agents increases and the perturbation budget grows, our CP-Guard+ still maintains the highest scores, despite a slight performance degradation. For example, when $\Delta = 0.5$ and $N_{mal} = 2$, our CP-Guard+ achieves 63.64% AP@0.5 and 60.86% AP@0.7, which are 6.70% and 51.73% higher than ROBOSAC, and 0.76% and 39.66% higher than MADE, respectively. These results demonstrate the robustness of our CP-Guard+ against malicious agents in CP systems, and show the superiority of our CP-Guard+ over existing CP defense methods.

**FPS Comparison.** We compare the FPS performance of CP-Guard+ with MADE and ROBOSAC, as shown in Figure 5 (a). The median FPS values for MADE, ROBOSAC, and CP-Guard+ are 56.86, 20.76, and 70.36, respectively. CP-Guard+ achieves a 23.74% higher FPS than MADE and a 238.92% increase over ROBOSAC, representing a significant improvement. These results highlight the high computational efficiency of our CP-Guard+.

### 5.3 ABLATION STUDY

**The Effect of Mixed Contrastive Training.** In this section, we evaluate the impact of mixed contrastive training on the performance of CP-Guard+. As shown in Figure 5 (c), we compare the performance of CP-Guard+ with and without this training strategy. The results demonstrate a significant performance improvement with mixed contrastive training. Specifically, Accuracy increases from 90.23% to 98.08%, TPR from 84.12% to 97.07%, Precision from 73.17% to 93.45%, and F1 score from 77.69% to 95.29%, with an average improvement of 19.06%. Additionally, Figure 5 (b) visualizes the cosine distance between intermediate features of malicious and benign agents. As training progresses, the cosine distance between negative pairs (benign and malicious features) increases, while the distance between positive pairs (benign-benign or malicious-malicious features) decreases. This indicates that mixed contrastive training effectively regularizes feature distribution, bringing positive pairs closer and separating negative pairs, as shown in Figure 4(b), thus enhancing the model's ability to differentiate between malicious and benign agents.

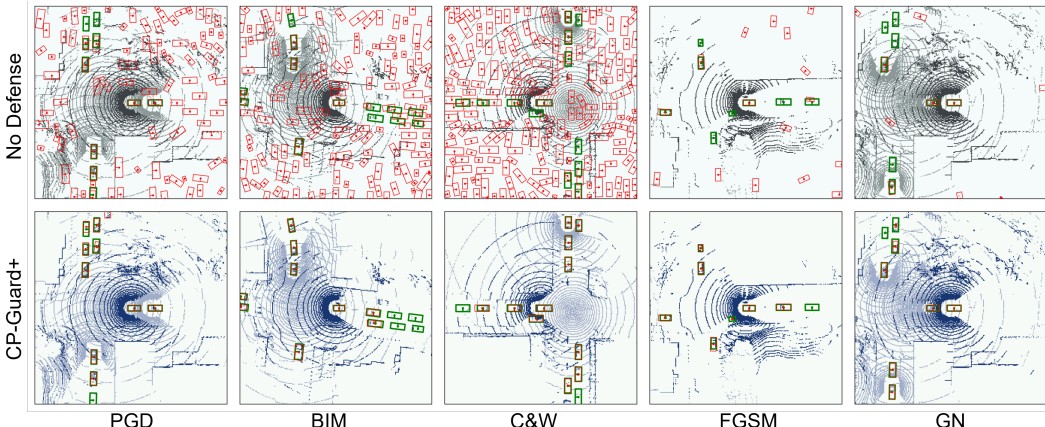

Figure 7: **Visualization and Qualitative Results.** We visualize the results of the CP systems with and without defense by CP-Guard+. The red bounding boxes represent the predicted outcomes, while the green ones denote the ground truth.

**The Impact of Perturbation Budget.** To assess the effect of perturbation budget on CP-Guard+ performance, we plot ROC curves for CP-Guard+ on CP-GuardBench with varying perturbation budgets $\Delta$ (0.1, 0.25, 0.5, 0.75, and 1.0), as depicted in Figure 6. The results show that when $\Delta = 0.1$, the area under the curve (AUC[1]) is 0.94, and as $\Delta$ increases, the AUC also increases. Specifically, when $\Delta$ reaches 0.5, AUC approaches 1, and at $\Delta = 1.0$, AUC nearly saturates at 1. This suggests that CP-Guard+ is more resilient to larger perturbation budgets, which is expected since larger perturbation budgets make malicious agents more discernible. This phenomenon is also evident in Table 1. For instance, when $\Delta = 0.25$, CP-Guard+ achieves an average accuracy of 95.78%, which increases to 98.53% at $\Delta = 1.0$. Overall, CP-Guard+ demonstrates robust performance across various perturbation budgets.

### 5.4 QUALITATIVE RESULTS

We visualize the results of the CP system with a malicious agent and the defense mechanism, CP-Guard+, as depicted in Figure 7. The red bounding boxes represent the predicted outcomes, while the green ones denote the ground truth. In the top row, which displays results without defense, malicious agents successfully blend into the crowd and mislead perception results, resulting in numerous false positive predictions. This significantly impacts the performance of CP systems and poses substantial security risks. Conversely, the bottom row showcases results with CP-Guard+. Here, malicious agents are effectively detected and eliminated, significantly reducing false positive predictions and increasing the true positive rate. These visualizations further confirm the effectiveness of CP-Guard+.

## 6 CONCLUSION

In this paper, we have proposed a new paradigm for malicious agent detection in CP systems, which directly detects malicious agents at the feature level without generating multiple hypothetical results but with significantly reduced system complexity and computation cost. We have also constructed a new benchmark, CP-GuardBench, for malicious agent detection in CP systems, which is the first benchmark in this field. Furthermore, we have developed CP-Guard+, a resilient method for detecting malicious agents in CP systems, which is capable of identifying malicious agents at the feature level without the need to verify the final perception results. Additionally, we have carefully designed a mixed contrastive training strategy to further fortify the resilience of CP-Guard+. Finally, we have conducted comprehensive experiments on V2X-Sim and our CP-GuardBench. The results have demonstrated the effectiveness and efficiency of CP-Guard+ in detecting malicious agents in CP systems.

---

[1]A higher AUC indicates better model performance.

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

# A  RELATED WORK

## A.1  COLLABORATIVE PERCEPTION

Collaborative Perception (CP) significantly extends the field-of-view (FoV) for individual agents, thereby enhancing the comprehensiveness and accuracy of perception outcomes (Han et al., 2023; Hu et al., 2024b). In CP systems, CAVs utilize various fusion methods tailored at distinct stages of data processing. Early fusion at the raw data level and late fusion at the output level often result in either high communication loads or increased perceptual noise. Conversely, intermediate fusion, which involves the transmission of intermediate features among CAVs, achieves an optimal balance by minimizing communication overhead while maximizing perceptual accuracy. Based on intermediate-level collaboration, recent progress in CP have addressed a wide array of challenges, including communication overhead (Fang et al., Aug. 2024; Tao et al., 2024), robustness (Lu et al., 2023), system heterogeneity (Lu et al., 2024), and domain generalization (Hu et al., 2023). Robustness, in particular, has become a pivotal area of focus, tackling issues such as communication disruptions (Ren et al., 2024), pose noise correction (Lu et al., 2023), and system latency (Lei et al., 2022). Despite comprehensive research, the vulnerability of these systems to malicious attacks has not been adequately addressed. In this paper, we delve into the robustness of CP systems, specifically considering the impact of malicious agents, and propose strategies to enhance system security and integrity.

## A.2  ADVERSARIAL COLLABORATIVE PERCEPTION

Adversarial attacks on single-vehicle perception systems typically utilize methods such as GPS spoofing (Li et al., 2021), LiDAR spoofing (Hallyburton et al., 2022), and deploying physically realizable adversarial objects (Tu et al., 2020). However, in multi-vehicle collaborative perception, adversarial strategies differ markedly across collaboration stages. For early-stage collaborative perception, Zhang *et al.* (Zhang et al., 2024) have identified attacks that involve object spoofing and removal, exploiting vulnerabilities through simulated object presence or absence and advanced reconstruction of LiDAR point clouds. Conversely, late-stage collaboration, which mainly involves sharing object locations (Schiegg et al., 2020), offers adversaries opportunities to manipulate these data points. Attacks at the intermediate stage are more complex, typically requiring white-box access to perception models. Such access allows attackers to precisely manipulate system outputs, though these systems are generally less vulnerable to simple black-box strategies like ray-casting due to the protective nature of benign feature maps that diminish the impact of these attacks. Pioneering work by Tu *et al.* (Tu et al., 2021) introduced untargeted adversarial attacks aimed at generating inaccurate detection bounding boxes by altering feature maps in intermediate-fusion systems. Building on this, Zhang *et al.* (Zhang et al., 2024) have enhanced these techniques by incorporating perturbation initialization and feature map masking, enabling more realistic and targeted attacks in real-time scenarios. Our research focuses on identifying and mitigating adversarial threats within the intermediate-level collaborative perception framework to bolster system resilience against these sophisticated attacks.

## A.3  DEFENSIVE COLLABORATIVE PERCEPTION

To enhance the resilience of intermediate-level CP against adversarial threats, contemporary research has primarily focused on the detection of malicious agents at the output level. Li *et al.* (Li et al., 2023) developed the Robust Collaborative Sampling Consensus (ROBOSAC) method that selects a random subset of collaborators for consensus verification. Additionally, Zhao *et al.* (Zhao et al., 2024) introduced match loss and reconstruction loss as metrics to assess consensus between an ego CAV and its collaborators' perception results for the detection of malicious agents. Furthermore, Zhang *et al.* (Zhang et al., 2024) utilized occupancy maps to identify inconsistencies between an ego CAV and other collaborators. In addition, our previous work, CP-Guard, which is currently under review, leverages the collaborative bird's eye view (BEV) segmentation results to iteratively check the normality from different collaborators to defend against malicious agents. However, these approaches adhere to a hypothesize-and-verify workflow, necessitating the generation of hypothetical perception outcomes and subsequent verification of their consistency with those of collaborators. This methodology is notably time-consuming and resource-intensive, hindering the system scalabil-

Table 3: Definitions of Evaluation Metrics

| Metric | Definition | Range |
|---|---|---|
| Accuracy | (TP + TN) / (TP + TN + FP + FN) | $[0, 1]$ |
| True Positive Rate (TPR) | TP / (TP + FN) | $[0, 1]$ |
| False Positive Rate (FPR) | FP / (FP + TN) | $[0, 1]$ |
| Precision | TP / (TP + FP) | $[0, 1]$ |
| F1 Score | 2TP / (2TP + FP + FN) | $[0, 1]$ |
| Area Under the Curve (AUC) | Integral area of plotting TPR vs FPR | $[0, 1]$ |
| AP@0.5 | Average Precision at IoU=0.5 | $[0, 1]$ |
| AP@0.7 | Average Precision at IoU=0.7 | $[0, 1]$ |
| Frame Per Second (FPS) | Number of frames processed / Time (s) | $[0, \infty)$ |

ity. In this paper, we have proposed a novel approach that shifts the focus to detecting malicious agents at the feature level, thus circumventing the need to verify final perception results.

## B  DETAILS OF EVALUATION METRICS

In our experiments, we use Accuracy, True Positive Rate (TPR), False Positive Rate (FPR), Precision, F1 Score, AP@0.5, AP@0.7, and Frame Per Second (FPS) to evaluate the performance of our CP-Guard+. The definitions of these metrics are shown in Table 3.

## C  IMPLEMENTATION OF ATTACKS

We introduce five types of attacks in our paper, including Projected Gradient Descent (PGD), Carini & Wagner (C&W) attack, Basic Iterative Method (BIM), Fast Gradient Sign Method (FGSM), and Guassian Noise Perturbation (GN). The details of these attacks are as follows:

1. *Projected Gradient Descent (PGD)*: PGD is similar to BIM but with an additional random initialization step. The mathematical formulation for PGD is as follows:

$$\mathbf{F}_k^0 = \mathbf{F}_k + \text{Uniform}(-\Delta, \Delta) \tag{12}$$

$$\mathbf{F}_k^{t+1} = \Pi_\Delta \{ \mathbf{F}_k^t + \alpha \cdot \text{sign}(\nabla_{\mathbf{F}_k^t} \mathcal{L}(\mathbf{F}_k^t, \mathbf{y})) \} \tag{13}$$

where $t$ is the iteration index, $\alpha$ is the step size, $\epsilon$ is the maximum perturbation allowed, $\Pi_\Delta$ is the projection operation that ensures the perturbation is within the $\Delta$-ball of $\mathbf{F}_k$. The process is repeated for a fixed number of iterations $T$. In our implementation, we set $T = 15$ and $\alpha = 0.1$.

2. *Carini & Wagner (C&W) attack*: The C&W attack aims to find the smallest perturbation $\delta$ that can cause misclassification. It can be formulated as an optimization problem:

$$\min_\delta \|\delta\|_p + c \cdot f(\mathbf{F}_k + \delta) \tag{14}$$

where $\| \cdot \|_p$ is the $L_p$ norm, $c > 0$ is a constant, and $f$ is an objective function that encourages misclassification:

$$f(\mathbf{F}_k') = \max(\max_{i \neq t} Z(\mathbf{F}_k')_i - Z(\mathbf{F}_k')_t, -\kappa) \tag{15}$$

Here, $Z(\mathbf{F}_k')$ is the logit output of the model, $t$ is the target class, and $\kappa$ is a confidence parameter.

3. *Basic Iterative Method (BIM)*:

$$\mathbf{F}_k^{t+1} = \text{Clip}_\Delta \{ \mathbf{F}_k^t + \alpha \cdot \text{sign}(\nabla_{\mathbf{F}_k^t} \mathcal{L}(\mathbf{F}_k^t, \mathbf{y})) \} \tag{16}$$

where $t$ is the iteration index, $\alpha$ is the step size, $\Delta$ is the maximum perturbation allowed, $\text{Clip}_\Delta$ is a function that clips the values to be within the $\Delta$-neighborhood of the original features $\mathbf{F}_k$, and $\mathbf{F}_k^0 = \mathbf{F}_k$. The process is repeated for a fixed number of iterations or until a stopping criterion is met.

4. *Fast Gradient Sign Method (FGSM)*: FGSM generates adversarial examples by perturbing the input in the direction of the gradient of the loss function with respect to the input. Given the intermediate features $\mathbf{F}_k$, the mathematical formulation of FGSM is as follows:

$$\mathbf{F}_k^{\text{adv}} = \mathbf{F}_k + \Delta \cdot \text{sign}(\nabla_{\mathbf{F}_k} \mathcal{L}(\mathbf{F}_k, \mathbf{y})) \tag{17}$$

where $\mathbf{F}_k^{\text{adv}}$ is the adversarial example, $\Delta$ is the perturbation magnitude, $\mathcal{L}$ is the loss function, and $\mathbf{y}$ is the true label. The $\text{sign}(\cdot)$ function takes the sign of the gradient, ensuring that the perturbation is in the direction that maximizes the loss.

5. *Guassian Noise Perturbation (GN)*: This attack suits the scenario where an attacker has no information about the victim's model. It means that the attacker can only launch black-box attacks. In this attack, the attacker generates Gaussian noise $\delta_{\text{GN}}$ and perturbs the original features $\mathbf{F}_k$.

