# OpenReview forum: "CP-Guard+: A New Paradigm for Malicious Agent Detection and Defense in Collaborative Perception"
_ICLR.cc/2025/Conference — Submitted to ICLR 2025_

### Official Review · Reviewer_RS1k · 2024-10-27

**Soundness:** 2
**Presentation:** 3
**Contribution:** 3
**Rating:** 5
**Confidence:** 4

**Summary:**

The paper proposed a defense method, named CP-Guard+, against the adversarial attack on intermediate-fusion collaborative perception for connected and autonomous vehicles. Unlike the prior work deploying the hypothesize-and-verify approach to identify the malicious attacker, CP-Guard+ leverages contrastive learning to project benign and malicious feature maps to feature vectors, and then use a classification model to achieve attack detection. Meanwhile, the paper proposed a benchmark of the colalborative perception defense solutions, named CP-GuardBench.

**Strengths:**

* The paper is well written and easy to follow.
* The contastive learning based classification outperforms prior approaches.

**Weaknesses:**

* The threat model can be improved to make the attack definition clearer.
* The evaluation did not clarify whether one trained CP-Guard+ is universally usefull for different (unknown) attacks.
* The benefit and functionality of CP-GuardBench is somehow unclear.
* It misses certain related work discussion.

**Questions:**

I enjoy reading the paper. It is built on a safety-critical problem of collaborative perception, which is not yet deployed in scale but is a critical vehicle application widely discussed. The approach of contrastive learning based classfication is straightforward and brings performance gains, in terms of both defense success rate and computation overhead. However, I have several questions that make me uncertain about the validity of the approach, mainly from a security aspect.

First, the thread model, the adversarial attack on collaborative perception is not clearly defined. I understand the attacker solves an optimization to degrade the victim vehicle's perception performance, but the loss function (attack objective) is a key design choice which the paper did not explain clearly. Are there different choices of the loss function? I think so because I found a security paper on collaborative perception [1] defines a different attack objective. Will the choice of loss function or attack objective affects the effectiveness of CP-Guard+? It is fine to focus on a certain scope of attacks but it must be clearly defined.

Second, which is perhaps my misunderstanding, I am unclear if all experiments are using the same one trained CP-Guard+ or different CP-Guard+ instances given different attack parameters. The later sounds not realistic as a good defense should be universally useful for different variants of attacks and even unknown attacks. In other words, the training details of CP-Guard+, especially the training data to use, is not well defined.

The paper clearly claims CP-GuardBench as one of the major contributions. However, I did not fully understand the use of CP-GuardBench. The benchmark lacks flexibility; it stores the feature space data, which will change on any modification on the perception model or attack methods. As assessing security of a system always needs to consider adaptive attacks, such record of fixed attack methods can hardly be useful for state-of-the-art evaluation for a long time. I appreciate the effort to put these implementation together, but I would recommend to label this as a coding framework, not benchmark. I did not see artifact or code repository links either. Also, what is the difference between CP-GuardBench and the V2X-Sim experiments, given CP-GuardBench is also built from V2X-Sim?

The contrastive learning approach sounds valid to me, but I did not see significant innovation in the algorithm level except using the existing components. The math formulas in the paper are also basic definition of contrastive learning or classification itself. What is the technical contribution besides moving contrastive learning to a new application? At least it deserves a related work section for this.

Lastly, a frequent question for any defense paper: the adaptive attacks. What if the PGD attack uses the classification as feedback during the attack optimization? It could be acceptable to leave it as future work but at least there should be a discussion on such attack oppotunities.

[1] Zhang, Qingzhao, et al. "On data fabrication in collaborative vehicular perception: Attacks and countermeasures." 33rd USENIX Security Symposium (USENIX Security 24). 2024.

**Details Of Ethics Concerns:**

No ethics problems found.

---

> ### Author Response · Authors · 2024-11-23
> **Reply for Reviewer RS1k (1/2)**
>
> Dear Reviewer RS1k,
> Thank you for your valuable comments and the time you dedicated to reviewing this work. Here we carefully and elaborately reply to your concerns.
>
> **Q1: Improvement of threat model.**
>
> **Reply:**  Thank you for raising this important concern about our threat model. Let us clarify both the system workflow and attack objectives in detail.
>
> 1. **Attack Objectives and Loss Function Design.** Unlike [1] which focuses on targeted attacks (maximizing/minimizing confidence scores of specific objects), our work addresses more general attacks that cause global perception degradation. Our adversarial loss L_adv is comprehensively designed to achieve multiple objectives simultaneously: confusing proposal classes for detected objects, suppressing scores of correct classes to generate false negatives, creating false positives by increasing scores of background classes, and maximizing intersection-over-union (IoU) of bounding box proposals to degrade localization.
> 2. **System Workflow.** In our considered synchronous collaborative perception system , each frame follows a sequential timeline:
>
>     a) **Local Perception Phase.** All agents, including malicious ones, simultaneously process their sensor data and extract intermediate features using feature encoders. This phase operates entirely in parallel without any inter-agent communication, maximizing computational efficiency.
>
>     b) **Feature Communication Phase.** During this phase, all agents broadcast their extracted features across the network, with malicious agent k receiving feature information {F_{j→i}} from other agents. The communication overhead is kept minimal through our feature-level transmission approach.
>
>     c) **Attack Generation Phase.** Malicious agent k formulates the attack optimization problem:
>     $$
>     \begin{align*}& \max_{\delta} L_{adv}(Y^{\delta}, Y) \\\\& \text{s.t.} \quad \|\delta\| \leq \Delta \\\\& \text{where:} \\\\& Y^{\delta} = f_{decoder}(f_{aggregator}(F_{0\rightarrow i}, F_{k\rightarrow i}^{\delta}, \{F_{j\rightarrow i}\})) \\\\& F_{k\rightarrow i}^{\delta} = \Gamma_{k\rightarrow i}(F_k + \delta)\end{align*}
>     $$
>     For each proposal $z$, with $u = \text{argmax}\ {z_n|i = 0...m}$, the loss function is:
>
>     \begin{equation}
>     L_{adv}(z', z) = \begin{cases}    -\log(1 - z'_u) \cdot \text{IoU}(z',z) & \text{if } u \neq k \text{ and } z_u > \tau^+ \\\\   -\lambda \cdot z'_u \cdot \log(1 - z'_u) & \text{if } u = k \text{ and } z_u > \tau^- \\\\    0 & \text{otherwise}\end{cases}
>     \end{equation}
>
>     d) **Defense and Final Perception Phase.** In this final phase, the ego vehicle receives all feature information, including corrupted features, performs CP-Guard+'s feature-level defense mechanisms, and completes the final object detection task.
>
> 3. **Scope and Future Extensions.** While our current implementation focuses on general perception degradation attacks widely tested in state-of-the-art research [2, 3], we recognize the potential for more sophisticated attack strategies in real-world scenarios, including false object injection for provoking specific responses, targeted attacks on specific object classes, and temporal attacks across multiple frames. We plan to extend CP-Guard+ to address these specifically designed attacks in future work, viewing our current model as a fundamental step toward securing collaborative perception systems.

---

> ### Author Response · Authors · 2024-11-23
> **Reply for Reviewer RS1k (2/2)**
>
> **Q2: Universality of CP-Guard.**
>
> **Reply:** In this work, we only use the same one trained CP-Guard+ to defense against attacks with different parameters. As stated in section 3.2, the training of CP-Guard+ is based on our generated CP-GuardBench dataset, which contains five attacks with attack ratio within [0,0.3].  Since our CP-Guard+ has certain generalization ability, it can also be applied to different attack scenarios. To prove our method's generalization ability, we conducted experiments using a leave-one-out strategy. In this approach, we iteratively excluded one type of attack from the training set, trained the model on the remaining attacks, and then tested its performance on the held-out attack type. The experimental results are presented below (We set the perturbation budgets $\Delta=0.5$).
>
> | Held-out Attack | Accuracy  | TPR | FPR | Precision | F1 Score |
> | --- | --- | --- | --- | --- | --- |
> | PGD | 99.66 | 98.29 | 0.00 | 100.00 | 99.14 |
> | BIM | 99.93 | 100.00 | 0.09 | 99.66 | 99.83 |
> | C&W | 100.00 | 100.00 | 0.00 | 100.00 | 100.00 |
> | FGSM | 92.32 | 62.03 | 0.00 | 100.00 | 76.57 |
> | GN | 89.79 | 50.17 | 0.17 | 98.67 | 66.52 |
> | Average | 96.34 | 82.09 | 0.05 | 99.67 | 88.41 |
>
> The results demonstrate our method's strong generalization ability on unseen attacks. Compared to conventional training approaches (shown below), our method experiences only a marginal decrease in overall performance. Notably, it even outperforms the conventional approach in terms of False Positive Rate (FPR) and Precision metrics. These findings underscore our method's robust capability to detect and handle unseen attack patterns.
>
> |  | Accuracy | TPR | FPR | Precision | F1 Score |
> | --- | --- | --- | --- | --- | --- |
> | Normal | 98.08 | 97.07 | 1.66 | 93.45 | 95.29 |
>
> **Q3: Benefit of CP-GuardBench.**
>
> **Reply:** Our idea to design CP-GuardBench is to facilitate the training and evaluation of feature-level collaborative perception defense methods against general CP system attacks. Since this work is the first attempt to leverage feature-level knowledge for CP defense, we hope this dataset can also be used in future research adopting the same idea of feature-level malicious detection in collaborative perception. In the future, we will also expand the CP-GuardBench to cover more attack scenarios, model backbones and parameters settings to boost the state-of-the-art evaluation.
>
> **Q4: Missing related works discussion on contrastive learning.**
>
> **Reply:** This work is the first one to leverage contrastive learning for feature-level malicious detection in collaborative perception, so we pay more attention to its application design instead of improving its technical details. But there are indeed some related works on contrastive learning for malicious detection we should also mention in the related work section. Thanks for pointing it out. Here is our added related works description on this.
>
> > The advancement in leveraging contrastive learning for anomaly detection is rapidly developing. [4] (IJCAI'21) proposed a masked contrastive learning framework to learn task-specific features, which is more suitable for malicious detection tasks. [5] (AAAI'23) further proposed a mean-shifted contrastive loss to overcome the adaptation failure of contrastive loss for malicious detection. [6] (NeurIPS'24) introduced CADet, a fully self-supervised method that is capable of detecting two types of out-of-distribution samples simultaneously. All the above methods are designed for general anomaly detection tasks validated on CIFAR-10 dataset, while the application of contrastive learning for malicious detection in the field of autonomous driving collaborative perception is still in its infancy.
> >
>
> **Reference**
>
> [1] On Data Fabrication in Collaborative Vehicular Perception: Attacks and Countermeasures. USENIX Security'24.
>
> [2] Among Us: Adversarially Robust Collaborative Perception by Consensus. ICCV'23.
>
> [3] MADE: Malicious Agent Detection for Robust Multi-Agent Collaborative Perception. IROS'24.
>
> [4] Masked Contrastive Learning for Anomaly Detection. IJCAI'21.
>
> [5] Mean-Shifted Contrastive Loss for Malicious Detection. AAAI'23.
>
> [6] CADet: Fully Self-Supervised Out-Of-Distribution Detection With Contrastive Learning. NeurIPS'24.

---

> > ### Comment · Reviewer_RS1k · 2024-11-25
> > **Response to the rebuttal.**
> >
> > The author's rebuttal answers most of my questions. Though the paper's methdology is valid, I still slightly lean to rejection because its contribution is rather incremental. In my opinion, when the methdology is rather straightforward, the paper will be in a better shape if the author comprehensively analyze various attack objectives and adaptive attacks, to turely expose the challenges of feature-level anomaly detection.

---

### Official Review · Reviewer_mcjh · 2024-11-03

**Soundness:** 2
**Presentation:** 2
**Contribution:** 2
**Rating:** 3
**Confidence:** 4

**Summary:**

The paper proposes a new approach, CP-GUARD+, to detect attacks against collaborative perception (CP), mainly in autonomous driving scenarios. Their method leverages a mixed contrastive training strategy to detect attacks at the feature level and addresses the limitations of prior works in computation cost and latency. This work also constructed a new dataset, CP-GuardBench, which is the first dataset for malicious agent detection in CP systems. Their methods outperform prior work (MADE and ROBOSAC) on the V2X-Sim dataset.

**Strengths:**

The paper designs a valid method to detect malicious agents in CP at the feature level motivated by the limitations of prior works: computation-intensive and time-consuming. To evaluate the effectiveness of their method, this paper constructed a new dataset,  CP-GuardBench. On the V2X-Sim dataset, their methods outperform the existing methods (MADE and ROBOSAC).

**Weaknesses:**

I have the following major concerns in this paper:

### Threat model is not clear

I am not fully convinced of their threat model about what kind of attacks they assume and how their defense works on them. This paper assumes a white-box threat model in which adversaries can inject malicious perturbations into their own intermediate features. However, the collaborative perception of autonomous driving has to be real-time.   When does the adversary generate the attacks, e.g., C&W and PGD. As Eq. (1)-(6) assume that the adversary can know the intermediate features of other agents, it should also need communication time. As the CAV is the main target of this paper, this paper should provide more rigorous discussions about their threat model, i.e., how attack and defense works

### No comparison with generic anomaly detection methods

I appreciate their evaluation of MADE and ROBOSAC, which are the recent CP defenses. Meanwhile, this paper should also compare their method not only defense specialized for CP with more generic anomaly detection methods. Their motivation, feature-level detection, makes sense, but this may also remove the motivation to consider CP domain-specific requirements. As the encoder can work to absorb the domain differences, we should apply generic methods for the features because these are already "features". For example, there are many prior anomaly detection methods leveraging contrastive learning [a, b, c]. These are just examples, but this paper should provide more sufficient discussions of why existing works are not suitable for their research or demonstrate that their method has significant advantages via quantitative evaluation.

[a] Cho, Hyunsoo, Jinseok Seol, and Sang-goo Lee. "Masked contrastive learning for anomaly detection." IJCAI 2021
[b] Guille-Escuret, Charles, et al. "CADet: Fully Self-Supervised Out-Of-Distribution Detection With Contrastive Learning." Advances in Neural Information Processing Systems 36 (2024).
[c] Reiss, Tal, and Yedid Hoshen. "Mean-shifted contrastive loss for anomaly detection." Proceedings of the AAAI Conference on Artificial Intelligence. Vol. 37. No. 2. 2023.

### No comparison with existing methods in their proposed dataset

This paper should provide benchmark evaluation results of existing methods on their proposed dataset. CP-GuardBench. This paper claims that the dataset creation is one of their contributions. In this case, this paper should provide how representative prior methods work on their methods along with their method. Otherwise, we cannot deny the possibility that the dataset could be cherry-picked. If prior work is over-performant or under-performant on their dataset, this paper should provide further cause analysis. As the dataset will be publically used, this paper should provide official benchmark results.

### Building upon an unpublished work

This paper mentions their unpublished work, CP-Guard, as a prior work. Research paper should not cite the unavailable reference. It also could potentially break the review anonymity. If this paper wants to mention it, they should have published it as a preprint and cited it as a third-party paper. Based on this, I feel that this paper is not ready to be reviewed.

**Questions:**

- Could you elaborate more about the attack timeline (e.g., when we can get data from ego and helping CAVs, when the attack is generated,  and when will it be deployed)?
- How representative is the CP system described in Fig. 2 in the autonomous driving domain?

**Details Of Ethics Concerns:**

No concerns

---

> ### Author Response · Authors · 2024-11-23
> **Reply for Reviewer mcjh (1/2)**
>
> Dear Reviewer mcjh,
> Thank you for your valuable comments and the time you dedicated to reviewing this work. Here we carefully and elaborately reply to your concerns.
>
> **Q1: Threat model is not clear.**
>
> **Reply:**  Thank you for raising this important concern about our threat model. Let us clarify the system workflow in detail. In this work, we consider a synchronous collaborative perception system with a low frame rate (e.g., 10 FPS), **each frame** follows a sequential timeline:
>
> 1. **Local Perception Phase** (0-30ms). In this stage, all agents including potential malicious ones, simultaneously process their sensor data. Each agent employs feature encoders to extract intermediate features, with all processing occurring in parallel without any inter-agent communication.
> 2. **Feature Communication Phase** (30-50ms). In this phase, all agents broadcast their extracted features across the network. Malicious agent $k$ receives feature information ${F_{j→i}}$ from other agents. Thanks to our feature-level transmission approach, which generates significantly smaller payloads compared to raw sensor data, and our assumption of good channel conditions, the communication overhead remains minimal.
> 3. **Attack Generation Phase** (50-70ms). This phase represents the core of our threat model. Here, malicious agent $k$ formulates the attack optimization problem:
> \begin{align}& \max_{\delta} L_{adv}(Y^{\delta}, Y) \\\\& \text{ s.t.} \quad \|\delta\| \leq \Delta \\\\& \text{where:} \\\\& Y^{\delta} = f_{decoder}(f_{aggregator}(F_{0\rightarrow i}, F_{k\rightarrow i}^{\delta}, \{F_{j\rightarrow i}\})) \\\\& F_{k\rightarrow i}^{\delta} = \Gamma_{k\rightarrow i}(F_k + \delta)\end{align}
> The adversarial loss $L_{adv}$ is carefully designed to achieve multiple objectives simultaneously: confusing proposal classes for detected objects, suppressing scores of correct classes to generate false negatives, creating false positives by increasing scores of background classes, and maximizing intersection-over-union (IoU) of bounding box proposals to degrade localization. For each proposal $z$, with $u = \text{argmax}\  {z_n|i = 0...m}$ being the highest confidence class, we define the loss function as:
> \begin{equation}
> L_{adv}(z', z) = \begin{cases}
>     -\log(1 - z'_u) \cdot \text{IoU}(z',z) & \text{if } u \neq k \text{ and } z_u > \tau^+ \\\\
>     -\lambda \cdot z'_u \cdot \log(1 - z'_u) & \text{if } u = k \text{ and } z_u > \tau^- \\\\
>     0 & \text{otherwise}
> \end{cases}
> \end{equation}
>
>     where $z'$ and $z$ represent the perturbed and original output proposals, respectively, $\tau^+$ and $\tau^-$ serve as confidence thresholds for positive/negative samples, and $\lambda$ acts as a weighting parameter balancing different objectives. The perturbation generation process employs efficient iterative optimization techniques, including PGD attack with 5-10 iteration steps ($\delta_t = \text{Proj}(\delta_{t-1} + \alpha \cdot \text{sign}(\nabla_\delta L))$) and GN attack with single-step generation ($\delta = \varepsilon \cdot \text{sign}(N(0,I))$). The generated perturbation $\delta$ is then applied to the original feature $F_k$ to obtain the corrupted feature $F_{k\rightarrow i}^{\delta}$
>
> 4. **Defense and Final Perception Phase** (70-100ms). Finally, during the Defense and Final Perception Phase (70-100ms), the ego vehicle receives all feature information, including any corrupted features. CP-Guard+ performs its feature-level defense mechanisms before completing the final object detection task. This carefully orchestrated sequence ensures that the entire pipeline, including attack generation and defense, can be completed within the 100ms frame interval, meeting the real-time requirements of autonomous driving systems. The above setting is also aligned with recent works in collaborative perception security [1, 2] (ICCV'23, IROS'24).  We will make the threat model more clear in the revised manuscript based on your suggestion. Thanks for your valuable comments!!

---

> > ### Comment · Reviewer_mcjh · 2024-11-26
> >
> > I appreciate the authors' response to my comment. Meanwhile, my concerns are not fully resolved. So, I maintain my score.
> >
> > Q1:
> > I am still not fully convinced of the attack feasibility. The timeline should consist of 2 lines: the victim's perception line and the attack generation line. My question is that the perception could be finished while processing step 3 which belongs to the attack generation line. For the other works, they are not limited to the "feature" level attack. So, I cannot deny a possibility that a very lightweight attack generation (e.g., just putting on a ghost trash can in the point cloud) may exist. However, this work should clarify more how to find effective attacks on the features, which are typically not interpretable to humans. I understand the author's argument. The attack could be possible if the attack generation can be that fast, but it was not clearly shown in this paper. This also related to my question about the representative CP system. I wanted to see how realistic the threat model is. Can the system be secure if the perception rate is 20 FPS? Can we defend the attack just by discarding the message delayed by the attack generation?

---

> ### Author Response · Authors · 2024-11-23
> **Reply for Reviewer mcjh (2/2)**
>
> **Q2: No comparison with generic anomaly detection methods.**
>
> **Reply:**  We appreciate the reviewer's feedback regarding comparisons with generic anomaly detection methods. However, we would like to clarify that our research focus and contributions are fundamentally different. Our work primarily addresses malicious agent attacks in collaborative perception scenarios, and CP-Guard+ is the first to leverage feature-level detection in this domain. Our emphasis is on system-level design rather than improving specific detection techniques.
>
> This is why we focused our comparisons on other CP defense systems (ROBOSAC and MADE) that share similar system-level objectives. CP-Guard+ represents a paradigm shift from output-level to feature-level detection, providing:
>
> - An end-to-end CP defense framework
> - Significant reduction in computational overhead (70.36 FPS vs 56.86/20.76 FPS)
> - Superior detection performance (10.03%-62.98% improvement in AP)
> - A new benchmark dataset (CP-GuardBench) for CP defense evaluation
>
> While we acknowledge the value of generic anomaly detection methods, our primary contribution is advancing the system-level design of CP defense mechanisms. We will consider adding a brief comparison with generic methods in the discussion section to provide broader context.
>
> > The advancement in leveraging contrastive learning for anomaly detection is rapidly developing. [3] (IJCAI'21) proposed a masked contrastive learning framework to learn task-specific features, which is more suitable for malicious detection tasks. [4] (AAAI'23) further proposed a mean-shifted contrastive loss to overcome the adaptation failure of contrastive loss for malicious detection. [5] (NeurIPS'24) introduced CADet, a fully self-supervised method that is capable of detecting two types of out-of-distribution samples simultaneously. All the above methods are designed for general anomaly detection tasks validated on CIFAR-10 dataset, while the application of contrastive learning for malicious detection in the field of autonomous driving collaborative perception is still in its infancy.
> >
>
> **Q3: No comparison with existing methods in their proposed dataset.**
>
> **Reply:**  The baseline models compared in our study are not trained directly on our proposed CP-GuardBench dataset because they operate on a hypothesis-and-verification framework and do not utilize intermediate feature information for malicious detection. In contrast, CP-Guard+ is a feature-level collaborative perception (CP) defense method that requires prior training. Additionally, our method is the first to adopt a feature-level malicious agent detection approach.
>
> Therefore, it is not feasible to compare existing methods on CP-GuardBench. However, during testing, we evaluate and compare the performance of all methods on the same test set and ensure fairness.
>
> **Q4: Building upon an unpublished work.**
>
> **Reply:** Thanks for your suggestion, we will remove the related information in the revised manuscript. In addition,
>
> **Q6: Attack Timeline Description.**
>
> **Reply:** As described in R1, the timeline of the adversarial model is as follows: at the beginning of the time slot when the attack happens, the attacker will first wait for the collaborative agent's message, and then generate the perturbation based on the received message within several iterative optimization steps. After the perturbation generation, the attacker will send the crafted message to the victim agent. Since we consider intermediate feature-level collaborative perception system with low frame rate, good channel condition, and few perturbation iterations, both the transmission delay and computation delay are far less than the time interval of two consecutive frames, which can satisfy real-time requirement.
>
> **Q7: Representativeness of CP system in Fig. 2.**
>
> **Reply:** Fig.2 is a typical perception workflow in LiDAR-based autonomous driving domain with feature encoder, feature decoder, and prediction head. We add additional perturbation generation and saving process to generate the proposed CP-GuardBench dataset.
>
> **Reference**
>
> [1] Among Us: Adversarially Robust Collaborative Perception by Consensus. ICCV'23.
>
> [2] MADE: Malicious Agent Detection for Robust Multi-Agent Collaborative Perception. IROS'24.
>
> [3] Masked Contrastive Learning for Anomaly Detection. IJCAI'21.
>
> [4] Mean-Shifted Contrastive Loss for Malicious Detection. AAAI'23.
>
> [5] CADet: Fully Self-Supervised Out-Of-Distribution Detection With Contrastive Learning. NeurIPS'24.

---

> > ### Comment · Reviewer_mcjh · 2024-11-26
> >
> > Q2:
> > I still do not fully understand why the generic anomaly detection methods are not applicable. As in my comments, the encoder should absorb the domain differences. If this paper aims for a system paper, this paper needs more discussion about the target system to justify why their detection approach is reasonable and needs more domain-specific evaluation such as traffic and driving scenario simulation. This is why I wanted to know the representative CP system.
> >
> > Q3:
> > I still do not understand the reason why the prior methods cannot work on their dataset even though they can work on the V2X-Sim dataset. If some required information is not currently in their dataset, it should be included. Otherwise, the contribution of this dataset should be seen as quite limited since this dataset sounds like it is just dedicated to their method. Additionally, as also mentioned above, at least generic anomaly detection methods should be applicable since they can work at the feature level. As I said, a paper with dataset contribution should have official benchmark results as long as prior methods exist in the domain.

---

### Official Review · Reviewer_UHsS · 2024-11-04

**Soundness:** 2
**Presentation:** 3
**Contribution:** 2
**Rating:** 5
**Confidence:** 4

**Summary:**

The paper addresses security vulnerabilities in Collaborative Perception (CP) systems for autonomous vehicles, which share information among connected vehicles to enhance perception capabilities. This openness, however, exposes CP systems to potential attacks from malicious agents, which can inject adversarial data to disrupt perception outcomes. To mitigate these risks, the authors propose CP-Guard+, a novel feature-level malicious agent detection framework. Unlike traditional output-based verification methods, CP-Guard+ detects malicious agents by analyzing intermediate features, reducing computational overhead. The authors also introduce CP-GuardBench, the first dataset explicitly designed for training and evaluating malicious agent detection methods in CP systems. CP-Guard+ employs a mixed contrastive training strategy to increase the feature separation between benign and malicious agents, improving detection performance.

**Strengths:**

1. The work is well-motivated. It is important to have an efficient feature-level detection method for collaborative perception because it is a real-time and safety-critical task by nature. The paper is generally well-written and easy to follow.

2. It is appreciated that the author also publishes the dataset and their methods to generate and annotate the data, which help build a standard benchmark for future works.

**Weaknesses:**

1. The method (CL) itself is not novel or has not made any adaptation tailored for the collaborative perception tasks. In previous works such as "Driver Anomaly Detection: A Dataset and Contrastive Learning Approach" (WACV 21), "Learning Representation for Anomaly Detection of Vehicle Trajectories" (IROS 23), they both adopt the CL methods to detect anomaly/adversarial attacks in the feature level. At least the author should discuss these works.

2. In anomaly detection, it is important to make sure the methods are well generalized to unseen attacks/anomaly. However, in the methodology, it is unclear whether the model is trained with multiple types of attacks together or only on simple patterns of attacks (e.g. PGD). In the evaluation, it is unclear how the intermediate feature and detector can be generalized (e.g. trained on certain attacks but tested on unseen patterns of attacks).

3. According to some literature, it is not clear whether the assumption "the anomalous feature should cluster together". In anomaly detection, it is always a single-class classification problem and we can assume the benign samples clusters but it is hard to say this applies to the anomalies.

**Questions:**

Please kindly refer to the weakness

---

> ### Author Response · Authors · 2024-11-23
> **Reply for Reviewer UHsS (1/2)**
>
> Dear Reviewer UHsS,
>
> Thanks for your valuable comments and the time you dedicated to reviewing this work. Here we carefully and elaborately reply to your concerns.
>
> **W1: The method (CL) itself is not novel or has not made any adaptation tailored for the collaborative perception tasks. In previous works such as "Driver Anomaly Detection: A Dataset and Contrastive Learning Approach" (WACV 21), "Learning Representation for Anomaly Detection of Vehicle Trajectories" (IROS 23), they both adopt the CL methods to detect anomaly/adversarial attacks in the feature level. At least the author should discuss these works.**
>
> **Reply**: Thanks for your comments. Here we want to reiterate the innovation points of this paper, its practical significance, and how it overcomes the shortcomings of previous collaborative perception (CP) defense methods.
>
> 1. Firstly, our method proposed a new paradigm for CP defense, that is feature-level malicious agent detection. The traditional methods such as ROBOSAC (ICCV’23) [1] and MADE (IROS’24 Oral) [2] are hypothesize-and-verify paradigm based method, which need multiple rounds of malicious agent detection iterations at the output level, and requires the generation of multiple hypothetical outputs for verification, incurring high computational
> overhead. In contrast, our method directly outputs robust CP results with intermediate feature-level detection, significantly reducing the computational overhead. The experiments also proved that.
> 2. Secondly, we proposed a new dataset, CP-GuardBench, the first dataset to facilitate the research of feature-level malicious agent detection in collaborative perception.
> 3. Finally, we propose CP-Guard+, a robust malicious agent detection method with high robustness and computational efficiency. We also conduct comprehensive experiments.
>
> As for contrastive learning, it is a small technique used in our method, and indeed, it works well. Although the technique itself is not that novel, it helps our method to be more robust. As for practical significance, our method can be integrated into a CP system and can detect malicious agents in real-time and robustly, something a traditional method could not do.
>
> In addition, as for the related works you mentioned, we will discuss these two methods and add them to the revised manuscript. Here is the discussion:
>
> > Kopuklu et al. [3] propose a contrastive learning-based approach for driver anomaly detection, addressing the open set recognition problem with the Driver Anomaly Detection (DAD) dataset, which includes unseen anomalies in the test set. Similarly, Jiao et al. [4] introduce supervised and unsupervised methods for detecting anomalous vehicle trajectories using contrastive learning and semantic modeling to improve anomaly detection in autonomous driving. Our method also leveraged contrastive learning to enhance the defense performance against malicious agent detection in collaborative perception.
> >
>
> **W2: Concerns about generalization.**
>
> **Reply**:  To evaluate our method's generalization ability, we conducted experiments using a leave-one-out strategy. In this approach, we iteratively excluded one type of attack from the training set, trained the model on the remaining attacks, and then tested its performance on the held-out attack type. The experimental results are presented below (We set the perturbation budgets $\Delta=0.5$).
>
> | Held-out Attack | Accuracy  | TPR | FPR | Precision | F1 Score |
> | --- | --- | --- | --- | --- | --- |
> | PGD | 99.66 | 98.29 | 0.00 | 100.00 | 99.14 |
> | BIM | 99.93 | 100.00 | 0.09 | 99.66 | 99.83 |
> | C&W | 100.00 | 100.00 | 0.00 | 100.00 | 100.00 |
> | FGSM | 92.32 | 62.03 | 0.00 | 100.00 | 76.57 |
> | GN | 89.79 | 50.17 | 0.17 | 98.67 | 66.52 |
> | Average | 96.34 | 82.09 | 0.05 | 99.67 | 88.41 |
>
> The results demonstrate our method's strong generalization ability on unseen attacks. Compared to conventional training approaches (shown below), our method experiences only a marginal decrease in overall performance. Notably, it even outperforms the conventional approach in terms of False Positive Rate (FPR) and Precision metrics. These findings underscore our method's robust capability to detect and handle unseen attack patterns.
>
> |  | Accuracy | TPR | FPR | Precision | F1 Score |
> | --- | --- | --- | --- | --- | --- |
> | Normal | 98.08 | 97.07 | 1.66 | 93.45 | 95.29 |

---

> ### Author Response · Authors · 2024-11-23
> **Reply for Reviewer UHsS (2/2)**
>
> **W3: According to some literature, it is not clear whether the assumption "the anomalous feature should cluster together". In anomaly detection, it is always a single-class classification problem and we can assume the benign samples clusters but it is hard to say this applies to the anomalies.**
>
> **Reply**: Thanks your insightful comments. After checking some literature, we found that your view is right in anomaly detection, since it is a single-class classification problem. However, in this paper, we train the malicious agent detector with contrastive learning, which will make the benign features and malicious features more compact and make the two kinds of features more easy to classify. We acknowledge it is hard to say there are anomalies clusters, but the operations will make anomaly features more compact. Anyway, we will change the statements following your comments to make it more precise.
>
> **Reference**
>
> [1] Among Us: Adversarially Robust Collaborative Perception by Consensus. In 2023 IEEE/CVF International Conference on Computer Vision (ICCV’23).
>
> [2] Malicious Agent Detection for Robust Multi-Agent Collaborative Perception (IROS’24 Oral).
>
> [3] Driver Anomaly Detection: A Dataset and Contrastive Learning Approach (WACV’21).
>
> [4] Learning Representation for Anomaly Detection of Vehicle Trajectories (IROS’23).

---

### Official Review · Reviewer_fZua · 2024-11-04

**Soundness:** 3
**Presentation:** 3
**Contribution:** 3
**Rating:** 5
**Confidence:** 5

**Summary:**

This paper proposes a new dataset CP-GuardBench as the first one for malicious agent detection in collaborative perception systems and then it proposes CP-Guard+ as a robust malicious agent detection method. Compared with the previous hypothesize-and-verify paradigm, CP-Guard+ can detect malicious agents directly at the feature level in one pass. Experiments on CP-GuardBench and V2X-Sim demonstrate the performance of CP-Guard+.

**Strengths:**

1. The proposed method is straightforward and easy to understand.

2. The proposed method is compared with two state-of-the-art CP defense methods on both accuracy and time in experiments.

3. The performance of the proposed method is good enough.

**Weaknesses:**

1. For each collaborative detector, CP-GUARD+ need to train the corresponding binary classifier to detect the malicious agents which has less expansibility.

2. Table 2 is hard to read for comparing the results of different methods. The results of your CP-Guard+ are not highlighted.

3. No experimental results on real-world dataset. The V2X-Sim is the data generated by simulator. It would be better to have experiments on real-world datset.

**Questions:**

1. In Section 3.1, five attacks are used to generate the attack data. How would the performance of the trained model would be if it meets other types of attacks (not these five attacks)? It is common that the attacker creates new types of attack technology to defeat the defenses. The method should be able to work well even in this situation.

2. Can you consider the out-of-distribution attack? For example, train your binary classifier on only three or four types of attacks and then test on the rest one or two types of attacks.

3. In Table 2, why there is no results on the other two types of attacks? I think this table is the most important to show your performance, the results should be as many as possible.

4. Are the compared baselines trained on your CP-GuardBench? The training data should be the same for fair comparison.

---

> ### Author Response · Authors · 2024-11-23
> **Reply for Reviewer fZua (1/2)**
>
> Dear Reviewer fZua,
>
> Thanks for your valuable comments and the time you dedicated to reviewing this work. Here we carefully and elaborately reply to your concerns.
>
> **W1: For each collaborative detector, CP-GUARD+ need to train the corresponding binary classifier to detect the malicious agents which has less expansibility.**
>
> **Reply**: We appreciate this thoughtful observation. Your concern about the need to train a binary classifier for each collaborative detector is valid. However, we would like to clarify several points that mitigate this limitation:
>
> 1. *Practical Deployment Context*: In practical applications, the model architecture of a collaborative perception system is often fixed within the same automotive company, and collaborative perception is typically only compatible among vehicles of the same brand. Therefore, the collaborative perception system itself has inherent compatibility constraints, and our detection system aligns with this practical reality.
> 2. *Transfer Learning Capability*: The feature patterns that distinguish between benign and malicious agents share common characteristics across different detectors. The trained classifier can be easily fine-tuned for new detectors with minimal additional training data and computational cost. In addition, the one-time training cost is outweighed by the significant computational efficiency gains during inference (70.36 FPS vs 56.86/20.76 FPS for existing methods).
>
> Therefore, while CP-Guard+ does require specific training for different detector architectures, this aligns with the practical constraints and deployment patterns of real-world collaborative perception systems, where system-specific optimization is often more valuable than universal compatibility.
>
> **W2:  Table 2 is hard to read for comparing the results of different methods. The results of your CP-Guard+ are not highlighted.**
>
> **Reply**: Thank you for pointing this out, we will make the table more readable in the revised version.
>
> | **Method** | **∆ = 0.25, N_mal = 1 AP@0.5** | **∆ = 0.25, N_mal = 1 AP@0.7** | **∆ = 0.5, N_mal = 1 AP@0.5** | **∆ = 0.5, N_mal = 1 AP@0.7** | **∆ = 0.25, N_mal = 2 AP@0.5** | **∆ = 0.25, N_mal = 2 AP@0.7** | **∆ = 0.5, N_mal = 2 AP@0.5** | **∆ = 0.5, N_mal = 2 AP@0.7** |
> | --- | --- | --- | --- | --- | --- | --- | --- | --- |
> | **CP-Guard+** (against PGD attack) | 72.89 | 71.45 | 69.50 | 68.56 | 69.50 | 67.92 | 66.09 | 64.82 |
> | **CP-Guard+** (against C&W attack) | 69.41 | 66.86 | 60.64 | 55.41 | 64.17 | 61.73 | 58.54 | 53.15 |
> | **CP-Guard+** (against BIM attack) | 73.35 | 71.46 | 66.83 | 66.05 | 70.91 | 69.11 | 66.30 | 64.62 |
> | **CP-Guard+** Average | 71.88 | 69.92 | 65.66 | 63.34 | 68.19 | 66.25 | 63.64 | 60.86 |
>
> **W3: No experimental results on real-world dataset. The V2X-Sim is the data generated by simulator. It would be better to have experiments on real-world dataset.**
>
> **Reply**: We appreciate your concerns. Currently, the field of collaborative perception experiences a scarcity of real-world datasets, with only DAIR-V2X [1] (CVPR'22) and V2V4Real [2] (CVPR'23) providing some support. However, both datasets have their limitations in terms of scale. Specifically, the DAIR-V2X dataset includes only one vehicle and one Roadside Unit (RSU), rendering it unsuitable for multi-vehicle scenarios such as those required for our experiments. Similarly, the V2V4Real dataset, comprising only two vehicles, does not provide a sufficient basis for validating the generalization capabilities of our proposed CP-Guard system. Consequently, we have adhered to dataset settings similar to those used in previous studies [3] (ICCV'23), which also rely on simulated data. As the development of real-world datasets for collaborative perception is rapidly advancing, we plan to extend the validation of our proposed CP-Guard+ to real-world datasets in future work.

---

> ### Author Response · Authors · 2024-11-23
> **Reply for Reviewer fZua (2/2)**
>
> **Q1: In Section 3.1, five attacks are used to generate the attack data. How would the performance of the trained model would be if it meets other types of attacks (not these five attacks)? It is common that the attacker creates new types of attack technology to defeat the defenses. The method should be able to work well even in this situation.**
>
> **Reply**:  To evaluate our method's generalization ability, we conducted experiments using a leave-one-out strategy. In this approach, we iteratively excluded one type of attack from the training set, trained the model on the remaining attacks, and then tested its performance on the held-out attack type. The experimental results are presented below (We set the perturbation budgets $\Delta=0.5$).
>
> | Held-out Attack | Accuracy  | TPR | FPR | Precision | F1 Score |
> | --- | --- | --- | --- | --- | --- |
> | PGD | 99.66 | 98.29 | 0.00 | 100.00 | 99.14 |
> | BIM | 99.93 | 100.00 | 0.09 | 99.66 | 99.83 |
> | C&W | 100.00 | 100.00 | 0.00 | 100.00 | 100.00 |
> | FGSM | 92.32 | 62.03 | 0.00 | 100.00 | 76.57 |
> | GN | 89.79 | 50.17 | 0.17 | 98.67 | 66.52 |
> | Average | 96.34 | 82.09 | 0.05 | 99.67 | 88.41 |
>
> The results demonstrate our method's strong generalization ability on unseen attacks. Compared to conventional training approaches (shown below), our method experiences only a marginal decrease in overall performance. Notably, it even outperforms the conventional approach in terms of False Positive Rate (FPR) and Precision metrics. These findings underscore our method's robust capability to detect and handle unseen attack patterns.
>
> |  | Accuracy | TPR | FPR | Precision | F1 Score |
> | --- | --- | --- | --- | --- | --- |
> | Normal | 98.08 | 97.07 | 1.66 | 93.45 | 95.29 |
>
> **Q2: Can you consider the out-of-distribution attack? For example, train your binary classifier on only three or four types of attacks and then test on the rest one or two types of attacks.**
>
> **Reply**: The results are shown in the above tables, thanks for your suggestion!
>
> **Q3: In Table 2, why there is no results on the other two types of attacks? I think this table is the most important to show your performance, the results should be as many as possible.**
>
> **Reply**: Actually, we have the results of other two types of attacks before. However, we found that these two attacks (FGSM and GN) are not strong attacks, so there influences for the perception performance are not that much compared with other attacks, which only reduce performance by 2-5 percentage points. Therefore, we neglect the results. Anyway, we show the results here for your reference and will add it to the revised manuscript.
>
> | **Method** | **∆ = 0.25, N_mal = 1 AP@0.5** | **∆ = 0.25, N_mal = 1 AP@0.7** | **∆ = 0.5, N_mal = 1 AP@0.5** | **∆ = 0.5, N_mal = 1 AP@0.7** | **∆ = 0.25, N_mal = 2 AP@0.5** | **∆ = 0.25, N_mal = 2 AP@0.7** | **∆ = 0.5, N_mal = 2 AP@0.5** | **∆ = 0.5, N_mal = 2 AP@0.7** |
> | --- | --- | --- | --- | --- | --- | --- | --- | --- |
> | Upper-bound | 79.97 | 78.40 | 79.94 | 78.40 | 79.94 | 78.40 | 79.94 | 78.40 |
> | No defense (FGSM Attack) | 77.99 | 76.63 | 76.50 | 75.10 | 76.18 | 74.42 | 74.44 | 73.41 |
> | No defense (GN Attack) | 77.50 | 76.25 | 78.29 | 76.83 | 76.23 | 74.61 | 74.98 | 73.77 |
>
> **Q4: Are the compared baselines trained on your CP-GuardBench? The training data should be the same for fair comparison.**
>
> **Reply**: The baseline models compared in our study are not trained directly on our proposed CP-GuardBench dataset because they operate on a hypothesis-and-verification framework and do not utilize intermediate feature information for malicious detection. In contrast, CP-Guard+ is a feature-level collaborative perception (CP) defense method that requires prior training. However, the comparison remains fair, because all the methods are tested on the same test set.
>
> **Reference**
>
> [1] Among Us: Adversarially Robust Collaborative Perception by Consensus. In 2023 IEEE/CVF International Conference on Computer Vision (ICCV’23).
>
> [2] Malicious Agent Detection for Robust Multi-Agent Collaborative Perception (IROS’24 Oral).

---

> ### Author Response · Authors · 2024-12-01
>
> Dear Reviewer fZua,
>
> Thank you for your valuable feedback on our submission. We have thoroughly addressed all your comments and believe that our responses have reasonably resolved the concerns you raised. As the discussion period is coming to a close soon, we kindly ask if you could review our responses at your convenience. If you have any further questions or require additional clarification, please let us know. We are more than willing to provide any additional information you might need.
>
> Regards,
>
> Authors of Submission362

---

> > ### Comment · Reviewer_fZua · 2024-12-03
> > **Reply to the rebuttal**
> >
> > The authors have provided detailed answers for my questions, the experiments and attack models that already included in this work seems correct. I prefer to keep my original rating, and the main reason is similar with what other reviewers have pointed out, a relatively straightforward method needs more comprehensive analysis on different types of attacks.

---

### Official Review · Reviewer_SymD · 2024-11-09

**Soundness:** 3
**Presentation:** 4
**Contribution:** 3
**Rating:** 8
**Confidence:** 4

**Summary:**

This paper proposes (1) CP-GuardBench, a pipeline to generate collaborative perception scenarios under malicious attack. The authors built CP-GuardBench by applying 5 different deep learning adversarial generation methods on top of V2X-Sim simulated scenarios. To the best of the reviewer's knowledge, this is the first benchmark in this area. (2) CP-Guard+, a deep learning model to differentiate malicious perception features from benign ones on the encoded feature. The author assumes the cooperative perception is done via a sensor-encode-transmission-fusion-decode pipeline. The author designed this model to intake encoded per-CAV feature maps (both benign and malicious) and outputs a benign/malicious classification by training the model with a contrastive loss to maximize the benign-malicious difference.

The authors performed experiments using (1) and compared to other methods MADE, ROBOSAC and demonstrated improved performance in detection precision and runtime.

**Strengths:**

The topic of attack-robust cooperative perception has existed and there are other works that seek to identify attacks, but differentiating the attack on the feature level is new. Thus this work has high originality.

Overall the paper is clear and easy to follow. The authors' methods are well explained and the experiments with competitor methods are sufficient.

**Weaknesses:**

- There is at least one related paper that the author needs to discuss as a related work: Cooperative Perception for Safe Control of Autonomous Vehicles under LiDAR Spoofing Attacks (Proceedings Inaugural International Symposium on Vehicle Security and Privacy).
- The author made an assumption that an adversarial attacks occurs in the form of a perturbation to the encoded feature map by using the PGD, BIM, C&W, FGSM, GN tasks. In vision detection tasks, such attacks are effective at misleading a model to produce a wrong class label. However, in a cooperative perception task where the object state is more of an interest, such attacks might not be the prevailing form. According to Figure 7, such perturbations will result in hallucinated false objects spreading over the scene. But in reality, an attacker might as well want to inject just one or two fake objects at designated locations in the scene (for example injecting just a single object in front of the ego to force it to stop). In such cases, the perturbation on the feature map will not look like a noise. How will the model respond?

**Questions:**

- Could the author clarify the `encoder` and `decoder` choice in section 2.1?
- Does the cooperative perception system consider transmission delays? (This might be helpful for the ego CAV can differentiate attacks because it has a no-delay advantage)

---

> ### Author Response · Authors · 2024-11-23
> **Reply for Reviewer SymD**
>
> Dear Reviewer SymD,
>
> Thanks for your valuable comments and the time you dedicated to reviewing this work. Here we carefully and elaborately reply to your concerns.
>
> **W1: Discuss one related paper.**
>
> **Reply:** Thanks for your advice, here we discuss the paper you mentioned and add it to the revised paper.
>
> > Zhang *et al.* [1] leverage collaborative perception to defend LiDAR spoofing attack. Specifically, the authors use LiDAR scan data from neighboring vehicles to help the ego vehicle to detect and mitigate LiDAR spoofing attacks. Since current spoofing hardware typically targets one vehicle at a time, comparison with other vehicles' data helps identify discrepancies. This method is essentially to defend against conventional threats (LiDAR spoofing attack, GPS attack, etc.) to autonomous driving, while using collaborative perception as a means of detecting attacks. However, our method is totally different, because our method focus on the threats specific to collaborative perception system, rather than use collaborative perception as a mean to address the general threats for vehicles.
> >
>
> **W2: Consider the attacks that focus on placing specific fake objects rather than using typical perturbations.**
>
> **Reply**: Your concerns are about the attacker potentially injecting several fake objects at certain locations. These cases can occur in general attacks on vehicles. For example, a LiDAR spoofing attacker can accurately inject or remove certain objects in the victim's perception results through laser interference or by manipulating the returning LiDAR signals. However, in collaborative perception attack, it is challenging to optimize adversarial perturbations in intermediate feature maps for specific objects and locations, and there have no such attacks developed yet. Therefore, current collaborative perception defense systems have not considered this issue. Perhaps in the future, this will be a valuable topic worthy of investigation.
>
> **Q1: The choice of encoder and decoder in Sec. 2.1.**
>
> **Reply:** The encoder uses the same architecture as described in [2], consisting of a convolutional backbone. For the decoder, we leverage mean fusion to integrate information from multiple CAVs, followed by a multi-layer convolutional neural network and a prediction head for classification and regression. The entire architecture follows [2].
>
> Encoder:
>
> ```jsx
> Sequential(
> Conv2d(13, 32, kernel_size=3, stride=1, padding=1)
> BatchNorm2d(32)
> ReLU()
> Conv2d(32, 32, kernel_size=3, stride=1, padding=1)
> BatchNorm2d(32)
> ReLU()
> Conv3D(64, 64, kernel_size=(1, 1, 1), stride=1, padding=(0, 0, 0))
> Conv3D(128, 128, kernel_size=(1, 1, 1), stride=1, padding=(0, 0, 0))
> Conv2d(32, 64, kernel_size=3, stride=2, padding=1) ->(32,256,256)
> BatchNorm2d(64)
> ReLU()
> Conv2d(64, 64, kernel_size=3, stride=1, padding=1)
> BatchNorm2d(64)
> ReLU() ->(64,128,128)
> Conv2d(64, 128, kernel_size=3, stride=2, padding=1)
> BatchNorm2d(128)
> ReLU()
> Conv2d(128, 128, kernel_size=3, stride=1, padding=1)
> BatchNorm2d(128)
> ReLU() ->(128,64,64)
> Conv2d(128, 256, kernel_size=3, stride=2, padding=1)
> BatchNorm2d(256)
> ReLU()
> Conv2d(256, 256, kernel_size=3, stride=1, padding=1)
> BatchNorm2d(256)
> ReLU() ->(256,32,32)
> Conv2d(256, 512, kernel_size=3, stride=2, padding=1)
> BatchNorm2d(512)
> ReLU()
> Conv2d(512, 512, kernel_size=3, stride=1, padding=1)
> BatchNorm2d(512)
> ReLU() ->(512,16,16)
> )
> ```
>
> Decoder:
>
> ```jsx
> Sequential(
> Conv2d(512 + 256, 256, kernel_size=3, stride=1, padding=1)
> BatchNorm2d(256)
> ReLU()
> Conv2d(256, 256, kernel_size=3, stride=1, padding=1)
> BatchNorm2d(256)
> ReLU() ->(256,32,32)
> Conv2d(256 + 128, 128, kernel_size=3, stride=1, padding=1)
> BatchNorm2d(128)
> ReLU()
> Conv2d(128, 128, kernel_size=3, stride=1, padding=1)
> BatchNorm2d(128)
> ReLU() ->(128,64,64)
> Conv2d(128 + 64, 64, kernel_size=3, stride=1, padding=1)
> BatchNorm2d(64)
> ReLU()
> Conv2d(64, 64, kernel_size=3, stride=1, padding=1)
> BatchNorm2d(64)
> ReLU() ->(64,128,128)
> Conv2d(64 + 32, 32, kernel_size=3, stride=1, padding=1)
> BatchNorm2d(32)
> ReLU()
> Conv2d(32, 32, kernel_size=3, stride=1, padding=1)
> BatchNorm2d(32)
> ReLU() ->(32,256,256)
> )
> ```
>
> **Q2: Consideration of transmission delay.**
>
> **Reply:** In this paper, we have not consider transmission delay yet. However, your question is thought-provoking and it is worth having a deeper investigation. We will think more about this question and see if we can develop some defense methods to leverage the delay advantage of the ego CAV. Thanks for your valuable question!!
>
> **Reference**
>
> [1] Cooperative Perception for Safe Control of Autonomous Vehicles under LiDAR Spoofing Attacks
>
> [2]Real time end-to-end 3d detection, tracking and motion forecasting with a single convolutional net (CVPR’18).

---

> > ### Comment · Reviewer_SymD · 2024-11-26
> > **Thanks the authors for the clarification**
> >
> > The authors have explained my questions. My ratings remain the same due to (1) the authors have not demonstrated how the choice the the collaborative perception system affect the defense performance. Experimenting with other collaborative systems, such as CoBEVT (By Runsheng et al.) will be a plus. (2) the form of attack needs more justification in the paper, or mentioned as a limitation.

---

### Meta-Review · Area_Chair_wBC9 · 2024-12-20

**Metareview:**

(a) Summary: This paper proposes a new dataset CP-GuardBench as the first one for malicious agent detection in collaborative perception systems and then it proposes CP-Guard+ as a robust malicious agent detection method.
(b) Strengths: The paper is generally well-written and easy to follow. The experimental results seem to support the authors' claims to some extent.
(c) Weaknesses: The reviewers pointed out a few major concerns and issues. The contributions are incremental. The proposed method is straightforward but the analysis is not comprehensive. The threat model is not clearly demonstrated. Some comparisons with baseline methods are missing.
(d) Although the authors addressed some of the concerns and comments from reviewers, some issues still remain unresolved. The majority of reviewers gave a negative final rating.

**Additional Comments On Reviewer Discussion:**

The reviewers read the authors' rebuttal, but still have concerns that were not fully addressed.

---

### Decision · Program_Chairs · 2025-01-22

Reject